# BooVAE: Boosting Approach for Continual Learning of VAE

**Evgenii Egorov** [*†]
University of Amsterdam
egorov.evgenyy@ya.ru

**Anna Kuzina** [*†]
Vrije Universiteit
av.kuzina@yandex.ru

**Evgeny Burnaev**
Skoltech, AIRI
e.burnaev@skoltech.ru

## Abstract

Variational autoencoder (VAE) is a deep generative model for unsupervised learning, allowing to encode observations into the meaningful latent space. VAE is prone to catastrophic forgetting when tasks arrive sequentially, and only the data for the current one is available. We address this problem of continual learning for VAEs. It is known that the choice of the prior distribution over the latent space is crucial for VAE in the non-continual setting. We argue that it can also be helpful to avoid catastrophic forgetting. We learn the approximation of the aggregated posterior as a prior for each task. This approximation is parametrised as an additive mixture of distributions induced by encoder evaluated at trainable pseudo-inputs. We use a greedy boosting-like approach with entropy regularisation to learn the components. This method encourages components diversity, which is essential as we aim at memorising the current task with the fewest components possible. Based on the learnable prior, we introduce an end-to-end approach for continual learning of VAEs and provide empirical studies on commonly used benchmarks (MNIST, Fashion MNIST, NotMNIST) and CelebA datasets. For each dataset, the proposed method avoids catastrophic forgetting in a fully automatic way.

## 1 Introduction

Variational Autoencoders (VAEs) (Kingma & Welling, 2013) are deep generative models used in various domains (Lee et al., 2017; Zhou et al., 2020). The VAE model consists of deep neural networks (DNNs): an *encoder* (inference network) and *decoder* (generative network). DNNs are known to reduce their quality on previously learned tasks when trained on data from a new task. Several directions to address this problem of *catastrophic forgetting* were suggested. But this phenomenon is mainly considered without attention to the specific properties of the VAE. We want to discuss current approaches to continual learning and formulate requirements for the ideal solution.

Dynamic architecture approach adds task-specific last layers (multi-heads) to encoder and decoder for each task (Rusu et al., 2016; Nguyen et al., 2018; Li & Hoiem, 2018). We suppose that practical applications of VAE require the common latent space, which is violated in multi-heads. Using multi-heads requires deciding which head to apply to the new data and when to expand the architecture. They reduce reuse of similarities between tasks. Hence, we suppose that the approach *should keep the static architecture* for both encoder and decoder.

The generative replay (Shin et al., 2017; Rao et al., 2019) uses a "teacher" generative model for generating "fake" data that mimics former training examples. Then the "student" model is trained on joint "fake" and new data. This approach is conceptually simple, model-agnostic and overcomes forgetting. However, these benefits come with a computational price. We need to retrain model while generating the dataset from all the past tasks, asses samples quality and the task-balance. Thus the approach *should avoid generative replay*).

---

[*]These authors contributed equally

[†]The work was done while the author was at the Skoltech, Moscow, 2019

35th Conference on Neural Information Processing Systems (NeurIPS 2021).

Weight penalty approach (Liu et al., 2018; Kirkpatrick et al., 2017) forms the trust-region around the optimum of the previous task for protecting parameters. This approach preserves the architecture and avoids generative replay. However, for DNNs a change in the weights is a poor proxy for the difference in the outputs (Benjamin et al., 2018). It is an even more critical issue for VAE model as it consists of a pair DNNs: encoder and decoder. Thus the approach *should link the data-space and the latent-space*.

We propose a novel continual learning approach for the VAEs. For each task, we expand current prior to get the approximation of an aggregated posterior over the whole data. We parametrise approximation as an additive mixture of distributions induced by encoder evaluated at trainable pseudo-inputs. These pseudo-inputs link the data-space and latent-space and help to memorise knowledge about past tasks. The problem of matching the aggregated posterior is ill-posed since we observe only its empirical version. As a solution, we use a greedy boosting-like approach with entropy regularisation. This method encourages components in the learned approximation to be diverse, which is essential as we aim at memorising the current task with the fewest components possible. The proposed approach is orthogonal to other methods mentioned above and can be applied in combination with them. Our main contributions are the following:

- We relate the approximation of optimal prior, the aggregated posterior and the continual learning task for VAE model. We find optimal additive perturbation in order to approximate optimal prior distribution. We derive the algorithm of effective approximation of the optimal prior for the continual learning framework.
- We use this result and present *Boosting Approach for Continual Learning of VAE* (BooVAE), a framework for training VAE models in the continual framework with static architecture.
- We empirically validate the proposed algorithm on commonly used benchmarks (MNIST, Fashion-MNIST, NotMNIST) and CelebA for disjoint sequential image generation tasks. The proposed generative model could be efficiently used in a generative replay for discriminative models. We train both generative and discriminative models incrementally, avoiding retraining the generative model for each task from scratch or storing several generative models. We provide code at `https://github.com/AKuzina/BooVAE`.

## 2 Background

**Variational Autoencoders (VAEs)**  Let $p_\theta(\boldsymbol{x}, \boldsymbol{z})$ be the joint distribution of observed variable $\boldsymbol{x} \in \mathbb{R}^D$ and hidden latent variable $\boldsymbol{z}$, with the distribution $\pi(\boldsymbol{z})$. Given the distribution $\boldsymbol{x} \sim p_e(\boldsymbol{x})$ we aim to find $\theta$ which maximizes the marginal log-likelihood $\mathbb{E}_{p_e(\boldsymbol{x})} \left[ \log \int p_\theta(\boldsymbol{x}, \boldsymbol{z}) d\boldsymbol{z} \right]$. Since the marginal likelihood is often intractable, we solve this optimization problem by the variational inference (Jordan et al., 1999). Variational autoencoders (VAE) (Kingma & Welling, 2013) amortize inference with $q_\phi(\boldsymbol{z}|\boldsymbol{x})$ as variational posterior (encoder) and $p_\theta(\boldsymbol{x}|\boldsymbol{z})$ as likelihood (decoder). The encoder and decoder are parameterized by neural networks with parameters $\phi, \theta$. Nets are optimized simultaneously via maximisation of the evidence lower bound (ELBO) objective:

$$\mathcal{L}(\theta, \phi, \lambda) \triangleq \mathbb{E}_{\substack{p_e(\boldsymbol{x}) \\ q_\phi(\boldsymbol{z}|\boldsymbol{x})}} (\log p_\theta(\boldsymbol{x}|\boldsymbol{z})\pi(\boldsymbol{z}) - \log q_\phi(\boldsymbol{z}|\boldsymbol{x})) \leq \mathbb{E}_{p_e(\boldsymbol{x})} \left[ \log \int p_\theta(\boldsymbol{x}, \boldsymbol{z}) d\boldsymbol{z} \right]. \quad (1)$$

Typically, instead of the density $p_e(\boldsymbol{x})$ we are given a dataset $\{\boldsymbol{x}_n\}_{n=1}^N$ and consider an empirical distribution $\frac{1}{N} \sum_{n=1}^N \delta_{\boldsymbol{x}_n}(\boldsymbol{x})$.

**Optimal Prior and Aggregated Posterior**  Hoffman & Johnson (2016); Goyal et al. (2017) discuss the choice of the prior distribution over latent space $\pi(\boldsymbol{z})$ and observe that default choice of the Gaussian prior significantly restricts the expressiveness of the model. Tomczak & Welling (2018) proposes to use a prior that optimizes the ELBO(1). The solution of this problem is the aggregated variational posterior:

$$\hat{q}(\boldsymbol{z}) = \mathbb{E}_{p_e(\boldsymbol{x})} q_\phi(\boldsymbol{z}|\boldsymbol{x}). \quad (2)$$

For an empirical $p_e(\boldsymbol{x})$, $\hat{q}(\boldsymbol{z}) = \frac{1}{N} \sum_{n=1}^N q_\phi(\boldsymbol{z}|\boldsymbol{x}_n)$ and Tomczak & Welling (2018) proposes a parametric approximation to avoid over-fitting and reduce computational costs: $\pi(\boldsymbol{z}) = \frac{1}{K} \sum_{k=1}^K q_\phi(\boldsymbol{z}|\boldsymbol{u}_k)$. Parameter $\{\boldsymbol{u}_k\}_{k=1}^K \in \mathbb{R}^{K \times D}$ is optimized simultaneously with $(\theta, \phi)$ by the ELBO(1) maximization, and $K$ is a hyper-parameter of the algorithm. The additive structure of the optimal prior $\pi^*(\boldsymbol{z})$ over the dataset points motivates us to consider such a prior distribution for continual learning.

**Continual learning framework**    In the continual learning framework, we do not have access to the whole dataset. We define the sequence of tasks to be solved $t = 1, \ldots, T$. Subsets of the data for each task $\mathcal{D}^1, \ldots, \mathcal{D}^T$ arrive sequentially and we have access only to the data of the current task. For each task $t$, we consider the empirical distribution $p_e^t(\boldsymbol{x}) = \frac{1}{|\mathcal{D}^t|} \sum\limits_{\boldsymbol{x}_n \in \mathcal{D}^t} \delta_{\boldsymbol{x}_n}(\boldsymbol{x})$.

# 3    BooVAE Algorithm (Proposed Method)

We start from defining an optimal prior for the VAE model in the continual learning framework. Next, we find the optimal additive expansion of the current prior to match the innovation coming from the new task. We use obtained result and provide a general algorithm for training the VAE model in the continual learning framework. It works as an iterative Minorization-Maximization algorithm. In the minorization step, we expand the current approximation of the prior and learn pseudo-inputs. At the Maximization step, we update parameters of the encoder and the decoder with the prior being fixed.

## 3.1    Optimal prior in continual learning

In this section, we derive an optimal prior for the VAE model in the continual framework and provide the algorithm to it approximation. We provide skipped technical details in Sup.(A.1). We start from the following decomposition of the ELBO(1):

$$\mathcal{L}(\theta, \phi, \pi) = \mathbb{E}_{p_e(\boldsymbol{x})}[\mathbb{E}_{q_\phi(\boldsymbol{z}|\boldsymbol{x})} \log p_\theta(\boldsymbol{x}|\boldsymbol{z}) - (\mathrm{KL}[q_\phi(\boldsymbol{z}|\boldsymbol{x})|\hat{q}(\boldsymbol{z})] + \mathrm{KL}[\hat{q}(\boldsymbol{z})|\pi(\boldsymbol{z})])], \tag{3}$$

where $\hat{q}(\boldsymbol{z}) = \mathbb{E}_{p_e(\boldsymbol{x})} q_\phi(\boldsymbol{z}|\boldsymbol{x})$ is the aggregated variational posterior. As KL-divergence is non-negative, the global maximum of Eq.(3) over $\pi$ is reached when:

$$\pi^*(\boldsymbol{z}) = \hat{q}(\boldsymbol{z}) = \mathbb{E}_{p_e(\boldsymbol{x})} q_\phi(\boldsymbol{z}|\boldsymbol{x}). \tag{4}$$

We assume that for the sequence of the two tasks, we can write the data distribution as the discrete mixture of two distributions: $p_e(\boldsymbol{x}) = \alpha\, p_e^1(\boldsymbol{x}) + (1-\alpha)\, p_e^2(\boldsymbol{x}), \alpha \in (0; 1)$. This assumption holds for empirical data distribution, which is of the most interest for us: $p_e(\boldsymbol{x}) = \frac{1}{N} \sum_{n=1}^{N} \delta_{\boldsymbol{x}_n}(\boldsymbol{x}) = \frac{|\mathcal{D}^1|}{|\mathcal{D}^1|+|\mathcal{D}^2|} p_e^1(\boldsymbol{x}) + \frac{|\mathcal{D}^2|}{|\mathcal{D}^1|+|\mathcal{D}^2|} p_e^2(\boldsymbol{x})$. Then we can decompose the ELBO(1) as following:

$$\mathcal{L}(\theta, \phi, \pi) = \mathbb{E}_{\substack{p_e(\boldsymbol{x}) \\ q_\phi(\boldsymbol{z}|\boldsymbol{x})}} \log p_\theta(\boldsymbol{x}|\boldsymbol{z}) - \sum_{i=1,2} \alpha_i (\mathbb{E}_{p_e^i(\boldsymbol{x})} \mathrm{KL}[q_\phi(\boldsymbol{z}|\boldsymbol{x})|\hat{q}_i(\boldsymbol{z})] + \mathrm{KL}[\hat{q}_i(\boldsymbol{z})|\pi(\boldsymbol{z})]), \tag{5}$$

where $\alpha_1 = \alpha, \alpha_2 = 1 - \alpha$ and $\hat{q}_i(\boldsymbol{z}) = \mathbb{E}_{p_e^i(\boldsymbol{x})} q_\phi(\boldsymbol{z}|\boldsymbol{x})$. Keeping only terms with dependence over the prior distribution, we conclude to the optimization problem over the probability density space $\mathcal{P}$:

$$\min_{\pi(\boldsymbol{z}) \in \mathcal{P}} \alpha\, \mathrm{KL}[\hat{q}_1(\boldsymbol{z})|\pi(\boldsymbol{z})] + (1 - \alpha)\, \mathrm{KL}[\hat{q}_2(\boldsymbol{z})|\pi(\boldsymbol{z})]. \tag{6}$$

Hence, if we can store the data for both tasks we can use the same optimal prior $\mathbb{E}_{\boldsymbol{x}} q_\phi(\boldsymbol{z}|\boldsymbol{x})$, where expectation is taken with respect to $\alpha\, p_e^1(\boldsymbol{x}) + (1-\alpha)\, p_e^2(\boldsymbol{x})$. However, in continual setting we don't have access to the $p_e^1(\boldsymbol{x})$, neither it is reasonable to store $\hat{q}_1(\boldsymbol{z})$. Thus after Task 1, we consider using the approximation of the aggregated variational posterior of the first task $\hat{q}_1^a(\boldsymbol{z}) \approx \hat{q}_1(\boldsymbol{z})$. This modification of the problem (6) leads to the following prior:

$$\pi^{1,2}(\boldsymbol{z}) = \alpha \hat{q}_1^a(\boldsymbol{z}) + (1-\alpha)\hat{q}_2(\boldsymbol{z}) = \arg\min_{\pi(\boldsymbol{z}) \in \mathcal{P}} \alpha\, \mathrm{KL}[\hat{q}_a(\boldsymbol{z})|\pi(\boldsymbol{z})] + (1-\alpha)\, \mathrm{KL}[\hat{q}_2(\boldsymbol{z})|\pi(\boldsymbol{z})]. \tag{7}$$

Next, we consider that the optimal prior for the first task $\pi^1(\boldsymbol{z})$ is a good start for a new prior after observing the new task. Hence, we propose using an additive expansions of the current prior to match innovation induced by the new task. Then the objective of the problem (7) transforms to:

$$\min_{h \in \mathcal{P}} \alpha\, \mathrm{KL}[\hat{q}_1^a(\boldsymbol{z})|(1-\beta)\pi^1(\boldsymbol{z}) + \beta h(\boldsymbol{z})] + (1-\alpha)\, \mathrm{KL}[\hat{q}_2(\boldsymbol{z})|(1-\beta)\pi^1(\boldsymbol{z}) + \beta h(\boldsymbol{z})]). \tag{8}$$

The optimization problem (8) is bi-convex over $h$ and $\beta$. We consider the Functional Frank-Wolfe algorithm (informally, "boosting") (Wang et al., 2015) for the corresponded objective:

$$\begin{aligned}
\mathcal{F}_{\alpha, \beta}[\hat{q}_1^a(\boldsymbol{z}), \hat{q}_2(\boldsymbol{z}); \pi^1(\boldsymbol{z}), h(\boldsymbol{z})] = \\
= \alpha\, \mathrm{KL}[\hat{q}_1^a(\boldsymbol{z})|(1-\beta)\pi^1(\boldsymbol{z}) + \beta h(\boldsymbol{z})] + (1-\alpha)\, \mathrm{KL}[\hat{q}_2(\boldsymbol{z})|(1-\beta)\pi^1(\boldsymbol{z}) + \beta h(\boldsymbol{z})]).
\end{aligned} \tag{9}$$

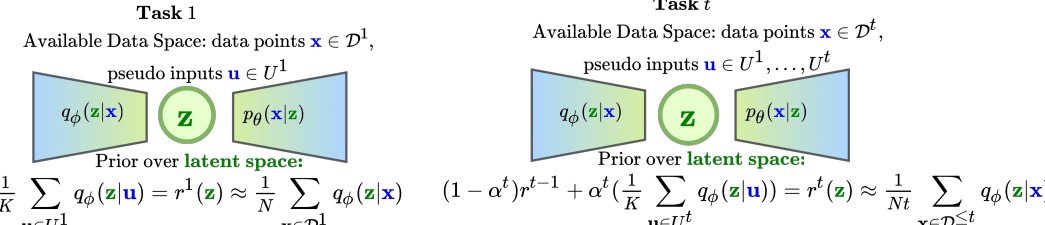

Figure 1: We propose to expand prior distribution in order to match new information from the coming task. We parametrize each component in the prior distribution with encoder, evaluated on the trainable pseudo-inputs. These pseudo inputs store information about the data at the correspondent task.

We aim to find $h$ as the projection of the functional gradient to the probability density space $\mathcal{P}$ and then optimize over $\beta$ with the stochastic gradient method, while $h$ fixed. The functional gradient of the objective (8) with respect to perturbation $h$ is $\alpha \frac{\hat{q}_1^a(z)}{\pi^1(z)} + (1-\alpha) \frac{\hat{q}_2(z)}{\pi^1(z)}$ in the following sense:

$$
\begin{aligned}
\mathcal{F}_{\alpha,\beta}[\hat{q}_1^a(z), \hat{q}_2(z); \pi^1(z), h(z)] = \\
= \mathcal{F}_\alpha[\hat{q}_1^a(z), \hat{q}_2(z); \pi^1(z)] - \beta \left( \int dz\, h(z) \left[ \alpha \frac{\hat{q}_1^a(z)}{\pi^1(z)} + (1-\alpha) \frac{\hat{q}_2(z)}{\pi^1(z)} \right] - 1 \right) + o(\beta).
\end{aligned}
\tag{10}
$$

We consider projection to the probability space as the solution of the optimization problem:

$$
\max_{h \in \mathcal{P}} \log \int h(z) \left[ \alpha \frac{\hat{q}_1^a(z)}{\pi^1(z)} + (1-\alpha) \frac{\hat{q}_2(z)}{\pi^1(z)} \right] dz \geq \max_{h \in \mathcal{P}} \int h(z) \log \left[ \alpha \frac{\hat{q}_1^a(z)}{\pi^1(z)} + (1-\alpha) \frac{\hat{q}_2(z)}{\pi^1(z)} \right] dz.
\tag{11}
$$

We use the lower bound in the (11) in order to use Monte-Carlo estimates of the gradients. The problem is linear over $h(z)$ and the solution is degenerate distribution. To this end, we add the entropy regularization $H[h] = -\int h(z) \log h(z)\, dz$ and obtain the final objective:

$$
\min_{h \in \mathcal{P}} \mathrm{KL} \left[ h \,\middle|\, \frac{\alpha \hat{q}_1^a(z) + (1-\alpha) \hat{q}_2(z)}{\pi^1(z)} \right].
\tag{12}
$$

As far as we update the initial prior $r^0(z) = \pi^1(z)$ with the mixture component $h$: $r^1(z) = (1-\beta)\pi^1(z) + \beta h(z)$, we can improve current approximation by finding the new component, solving the same optimization problem, but using $r^1$ instead of $\pi^1$. The objective (12) encourages a new component to be different from the already constructed approximation:

$$
\mathrm{KL} \left[ h \,\middle|\, \frac{\alpha \hat{q}_1^a(z) + (1-\alpha) \hat{q}_2(z)}{\pi^1(z)} \right] = \underbrace{\mathrm{KL} \left[ h | \alpha \hat{q}_1^a(z) + (1-\alpha) \hat{q}_2(z) \right]}_{\text{match the target prior}} - \underbrace{\int h(z) \log \frac{1}{\pi^1(z)} dz}_{\substack{\text{be different} \\ \text{from the current approximation}}}.
\tag{13}
$$

To this end, we have defined the algorithm of learning optimal prior for VAE in continual setting. Now we will use it to formulate the algorithm of VAE training in continual learning framework.

## 3.2 BooVAE Algorithm

In this section, we formulate the algorithm for continual learning – BooVAE (short for Boosting VAE). It works as the iterative Minorization-Maximization algorithm. In the *minorization step*, we use the obtained optimization problem (12) to learn a new component of the prior. These steps are alternated with *ELBO maximization steps* until the desired number of components in prior is reached. From that point further on, only model parameters are updated until convergence. As at maximization step follows the usual routine used in training VAEs, our approach can be easily used on top of any VAE model. The derivations of applications of BooVAE to the VAE with flow-based prior presented at Sup.(A.2). These steps are summarized in Alg.(1) and idea of prior learning for continual setting is illustrated in Fig.(1). Now, we are going to consider them in more details.

**Prior Update Step** At this step we expand the current approximation of the optimal prior distribution. Subsets $\mathcal{D}^1, \ldots, \mathcal{D}^T$ arrive sequentially and may come from different domains. We have access only to one subset of data and current prior at a time but aim at learning the prior distribution $\frac{1}{|\mathcal{D}^{\leq T}|} \sum_{\boldsymbol{x} \in \mathcal{D}^{\leq T}} q_\phi(\boldsymbol{z}|\boldsymbol{x})$. At the task $t$ we start from the current prior $r^t(\boldsymbol{z}) := r^{t-1}(\boldsymbol{z})$. We expand current approximation with at most $K$ components. At each step $k$ we add a new component $h$ to the current approximation $r^t$ to move it towards an optimal prior $\pi^{\leq t}$. Following (Tomczak & Welling, 2018), we select the parametric family $\mathcal{R}$ of the components $h$ as learnable pseudo-inputs $\boldsymbol{u}$ to the encoder: $h(\boldsymbol{z}) = q_\phi(\boldsymbol{z}|\boldsymbol{u})$. This choice connects parameters of the prior with the data-space. We conclude with a two step procedure for adding a new component:

1. Train new component:
$$h^* = \arg\min_{h \in \mathcal{R}} \mathrm{KL}\left[h \middle| \frac{\pi^{\leq t}}{r^t}\right].$$

2. Train component weight:
$$\beta^* = \arg\min_{\beta \in (0;1)} \mathrm{KL}\left[\beta h^* + (1-\beta)r^t \middle| \pi^{\leq t}\right].$$

---

**Algorithm 1** BooVAE algorithm

**input** Current task $t$ dataset $\mathcal{D}: \{x_n\}_{n=1}^N$
**input** Maximal number of components $K$
    Prior to approximate $\pi^{\leq t} = \alpha r^{t-1} + (1-\alpha)\hat{q}_t$.
    Initialize prior $r^t = r^{t-1}$
    $\theta^*, \phi^* = \arg\max_{\theta, \phi} \mathcal{L}(r^t, \theta, \phi)$
    $k = 1$
    **while** not converged and $k < K$ **do**
        $h^* = \arg\min_{h \in \mathcal{R}} \mathrm{KL}\left[h \middle| \frac{\pi_{\lambda^*}}{r^t}\right]$
        $\beta^* = \arg\min_{\beta \in (0;1)} \mathrm{KL}[\beta h + (1-\beta)r^t | \pi^{\leq t}]$
        $r^t = \beta^* h^* + (1-\beta^*)r^t$
        $k = k+1$
        $\theta^*, \phi^* = \arg\max_{\theta, \phi} \mathcal{L}(r^t, \theta, \phi)$
    **end while**
**output** $r^t, \theta^*, \phi^*$

---

As it was already mentioned, we define optimal prior to be equal to aggregate posterior. Therefore, it stores all the information about training dataset. The optimal prior for tasks $1:t$ can be expressed with prior from the previous step (e.g. aggregate posterior over tasks $1:t-1$) and training dataset from the current task only:

$$\pi^{\leq t}(\boldsymbol{z}) = \frac{|\mathcal{D}^{\leq t-1}|}{|\mathcal{D}^{\leq t}|} \pi^{\leq t-1}(\boldsymbol{z}) + \frac{|\mathcal{D}^t|}{|\mathcal{D}^{\leq t}|} \frac{1}{|\mathcal{D}^t|} \sum_{\boldsymbol{x_n} \in \mathcal{D}^t} q_\phi(\boldsymbol{z}|\boldsymbol{x_n}). \tag{14}$$

Since we don't have access to the data from the previous tasks, we suggest using trained prior $r^{t-1}$ as a proxy for the corresponding part of the mixture. We use random subset from the current task $\mathcal{M}^t \subset \mathcal{D}^t$ containing $N_t$ observations to estimate aggregate posterior of the current task.

$$\pi^{\leq t} \approx \frac{|\mathcal{D}^{\leq t-1}|}{|\mathcal{D}^{\leq t}|} r^{t-1}(\boldsymbol{z}|\{\boldsymbol{u}\}^{\leq t-1}) + \frac{|\mathcal{D}^t|}{|\mathcal{D}^{\leq t}|} \frac{1}{N_t} \sum_{\boldsymbol{x} \in \mathcal{M}^t} q_\phi(\boldsymbol{z}|\boldsymbol{x}). \tag{15}$$

Such formulation allows an algorithm not to forget information from the previous task, which is stored in the prior distribution and pseudo-inputs $\{\boldsymbol{u}\}^{\leq t-1}$. The prior distribution performs regularization for the VAE model. As the budget of components learning per-task is reached, we perform component pruning by weights optimization, see Supp.(A.3).

**ELBO Maximization Step.** At this step parameters of the encoder and decoder $\theta, \phi$ are updated. We add regularization term, to ensure that the model remembers the information, which is stored in the pseudo-inputs. Given the fixed mean-parameters of prior distribution from the previous task $r^{t-1}$, we keep its components during training the task $t$ as well as distribution of their fixed decoded mean-parameters $p^{t-1}(\cdot)$:

$$R_{enc}(\phi) = \sum_{\boldsymbol{u} \in r^{t-1}} \mathrm{KL}_{\mathrm{sym}}\left[q_\phi(\boldsymbol{z}|\boldsymbol{u})|r^{t-1}(\boldsymbol{z}|\boldsymbol{u})\right], R_{dec}(\theta) = \sum_{\boldsymbol{z} \sim r^{t-1}} \mathrm{KL}_{\mathrm{sym}}\left[p_\theta(\boldsymbol{x}|\boldsymbol{z})|p^{t-1}(\boldsymbol{x}|\boldsymbol{z})\right]. \tag{16}$$

where $\mathrm{KL}_{\mathrm{sym}}(p|q) = \frac{1}{2}\left(\mathrm{KL}(p|q) + \mathrm{KL}(q|p)\right)$. We use symmetric KL-divergence, since we have observed that it help to avoid encoder and decoder to memorizing delta-function centered in the pseudo-input and results in more diverse samples from the previous tasks. Final objective for the maximization step over $\phi, \theta$ for the task $t$ is the following:

$$\mathbb{E}_{\substack{\boldsymbol{x} \sim \mathcal{D}_t, \\ \boldsymbol{z} \sim q(\boldsymbol{z}|\boldsymbol{x})}} \left[\log p_\theta(\boldsymbol{x}|\boldsymbol{z}) - \mathrm{KL}[q_\phi(\boldsymbol{z}|\boldsymbol{x})|r^t(\boldsymbol{z})]\right] - \lambda(R_{enc}(\phi) + R_{dec}(\theta)). \tag{17}$$

# 4 Related Work

**Boosting density approximation.** The approximation of the unnormalized distribution with the sequential mixture models has been considered previously in several studies. Several works (Miller et al., 2017; Gershman et al., 2012) perform direct optimization of ELBO with respect to the new component's parameters. Unfortunately, it leads to unstable optimization problem. Therefore, other works consider functional Frank-Wolfe framework, where subproblems are linearized (Wang et al., 2015). At each step, KL-divergence functional is linearized at the current approximation point by its convex perturbation, obtaining tractable optimization subproblems over distribution space for each component. Guo et al. (2016) suggested using concave log-det regularization for gaussian base learners. Egorov et al. (2019), Locatello et al. (2018) used entropy-based regularization. In our work, we come up with a similar optimization algorithm. We optimize *different objective for different propose*: the weighted sum of exclusive KL to approximate optimal prior for VAE, while mentioned works approximate inclusive KL, in order to approximate posterior distriubtion.

**Continual learning for VAE.** (Lesort et al., 2019) compare EwC, generative replay and random coresets. We use a single pair of encoder and decoder for all the tasks unless otherwise stated. Nguyen et al. (2018) propose to use multi-head VAE architecture, where a separate encoder is trained for each task, and an extra head is added to the decoder. We believe that even though this model was shown to produce good results, it has several crucial limitations. Firstly, it makes scalability of the method very poor, since the number of parameters in the models increases dramatically with the number of tasks. Secondly, it requires task labels to be known both during training and validation to use the suitable "head" for each data point. Finally, it limits the amount of information we can share between tasks and thus may lead to worse performance for the tasks, which has fewer data available and have a lot of similarities with other tasks. We have observed the latter phenomenon on the CelebA dataset, where all the tasks are supposed to generate images with faces but with different hair color. Recent work (Rao et al., 2019) use learnable mixture as prior distribution. However, this reached by also expanding the architecture of encoder with multi-heads and using self-replay to overcome forgetting. Achille et al. propose similar ideas of encoder expansion and use also use self-replay. However, our goal is to provide orthogonal approach, which avoids self-replay and multi-heads, but can be combined with them.

# 5 Experiments

We empirically evaluation our algorithm on both disjoint image generation and classification tasks. In the latter case we suggest using VAE learned in the continual setting for generative replay (Shin et al., 2017).

## 5.1 Disjoint image generation task

**Setup.** We consider an experimental setup, where each task contains objects of 1 class (e.g. one digit for MNIST dataset). The resulting generative model, trained on all the classes one-by-one is supposed to generate images from all the classes. We perform experiments on MNIST, notMNIST, fashion MNIST and CelebA datasets. Each of MNIST datasets has 10 classes with digits, characters and pieces of clothing correspondingly. For the CelebA dataset we consider 4 tasks based on the attributes: black, blond, brown and gray hair. For all the datasets we use VAE (Kingma & Welling, 2013) model with gaussian posterior. A complete description of the experimental setup, architectures and hyperparameters can be found in Supp. (B.5).

**Compared methods.** As a baseline we consider standard VAE model. We expect it to suffer from catastrophic forgetting and use it as a lower bound on the model performance. We refer to this model as **Standard**. Our method has some similarities with the regularization-based continual learning algorithms, which add quadratic penalty on the weights. The difference is that we use prior, defined as a mixture of the encoded pseudo-inputs, to avoid catastrophic forgetting both for the encoder and the decoder, Eq. (16). In the experiments we compare with such regularization-based approaches as Elastic Weight Consolidation (**EWC**) (Kirkpatrick et al., 2017) and Variational Continual Learning (**VCL**) (Nguyen et al., 2018). Our approach uses pseudo-inputs to approximate an optimal prior distribution for each task. This procedure can also be seen as coreset selection (Huggins et al., 2016;

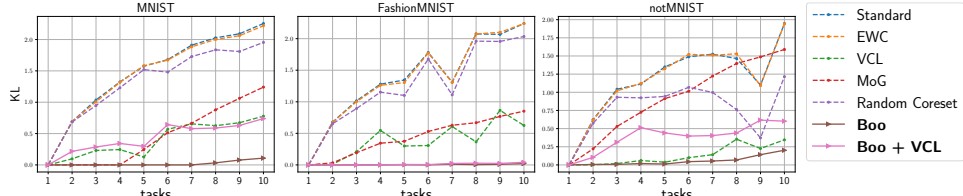

Figure 2: **Diversity** of generated samples, estimated as a KL between discrete distribution with the equal probability for each class and empirical distribution of samples from VAEs. The lower is the better. We conclude that BooVAE outperforms other approaches.

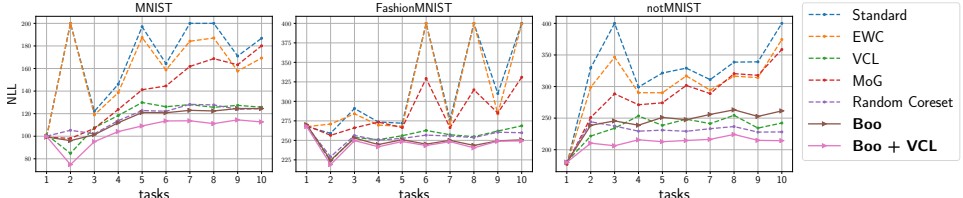

Figure 3: **NLL** on the full test dataset averaged over 5 runs after continually training on 10 tasks. Lower is better. We observe that BooVAE performance is comparable or better than of other methods.

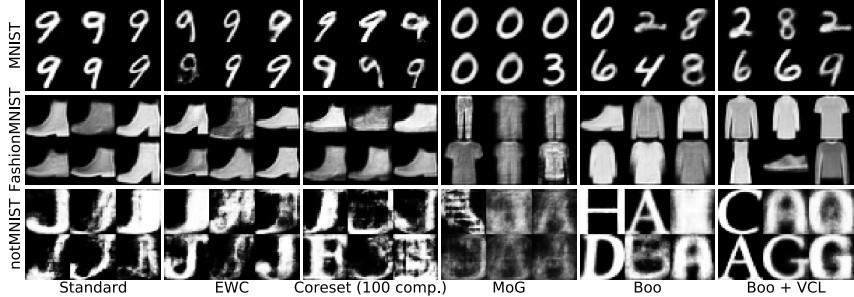

Figure 4: Samples from the VAEs after the last task is learned. BooVAE keep sampling different tasks, while other approaches suffer from catastrophic forgetting.

Bachem et al., 2015). To validate the quality of the learned coresets, we perform the comparison with **random coresets**, which was also used in (Nguyen et al., 2018). We store a random subsample of the training dataset from previous tasks and add them to each batch during training to avoid catastrophic forgetting. Moreover, we can learn the prior distribution components in the latent space instead of the data space as we do with the pseudo-inputs. We have conducted experiments, where the prior is trained as a mixture of Gaussian's (with diagonal covariance) in the latent space. We refer to this approach as **MoG**. Just as we do in BooVAE, in MoG we learn a new mixture of components for each task and use the regularization from Eq.(16) to avoid catastrophic forgetting.

**Metrics.** To evaluate the performance of the VAE approach on grey scale images, we a standard metric – negative log-likelihood (**NLL**) on the test set. NLL is calculated by importance sampling

Table 1: FID values for CelebA dataset averaged over 5 runs, the lower is the better. Each row corresponds to the FID for cumulative learned tasks of different hair types. BooVAE outperform other approaches, including multihead.

| #T | EWC | Multihead + EWC | VCL | Multihead + VCL | Random Coreset (40) | Random Coreset (80) | Boo (40 comp.) |
|---|---|---|---|---|---|---|---|
| 1 | 35.8 (0.8) | 37.7 (1.1) | 35.8 (0.6) | 37.93 (1.0) | 36.4 (1.3) | 38.1 (1.9) | **27.5 (0.6)** |
| 2 | 61.4 (1.7) | 69.2 (1.3) | 58.7 (0.6) | 65.54 (0.2) | 59.6 (0.4) | 59.8 (0.8) | **45.7 (2.4)** |
| 3 | 40.7 (0.6) | 45.2 (0.8) | 39.3 (0.6) | 42.18 (0.4) | 39.7 (0.8) | 41.9 (0.1) | **33.4 (1.4)** |
| 4 | 50.7 (0.6) | 75.7 (0.8) | 48.7 (0.9) | 75.16 (2.0) | 48.1 (1.1) | 47.4 (1.9) | **43.1 (0.2)** |

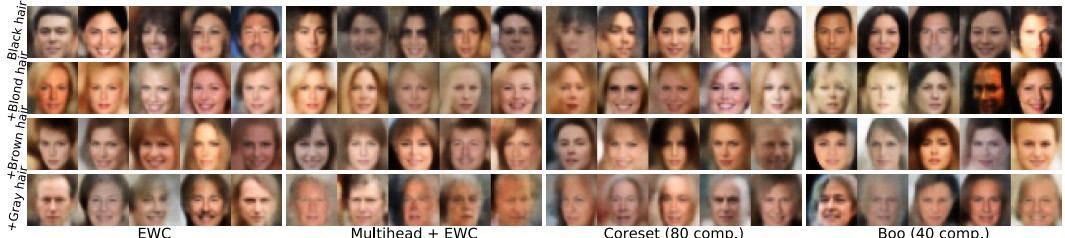

Figure 5: Samples from VAE trained on CelebA. For each model rows correspond to cumulatively learned tasks of different hair types.

method, using 5000 samples for each test observation:

$$-\log p(x) \approx -\log \frac{1}{K} \sum_{i=1}^{K} \frac{p_\theta(\boldsymbol{x}|\boldsymbol{z}_i)p(\boldsymbol{z}_i)}{q_\phi(\boldsymbol{z}_i|\boldsymbol{x})}, \ \boldsymbol{z}_i \sim q_\phi(\boldsymbol{z}|\boldsymbol{x}).$$

NLL measures the quality of the reconstructions that VAE produces. It is also essential to evaluate the diversity of generated images with respect to all the tasks in the continual setting. In our experiments, each task contains a new class. Thus, we expect our model to generate images from all the classes $t=1,\dots,T$ in the same proportion as they appear in the training dataset. For the dataset with balanced classes, this proportion is equal to $\frac{1}{T}$. We assess the diversity using the sum of **KL-divergences** between $T$ pairs of Bernoulli distributions:

$$\sum_{t=1}^{T} \mathrm{KL}\left[p_t||\widehat{p}_t\right], \ p_t \sim \mathrm{Be}\left(\tfrac{1}{T}\right), \ \widehat{p}_t \sim \mathrm{Be}\left(\tfrac{N_t}{\sum_{t=1}^{T} N_t}\right),$$

where $\widehat{p}_t$ is an empirical distribution of the generated classes, $N_t$ — number of generated images from the class $t$. To estimate $N_t$, we train the classification neural network to achieve high accuracy and use it to classify images generated by the model. We use $10^4$ samples in total to calculate the metric. For the CelebA dataset, we used Frechet Inception Distance (**FID**) (Heusel et al., 2017) over $10^4$ samples, which is supposed to measure both quality and diversity of generated samples. FID rely heavily on the implementation of Inception network (Barratt & Sharma, 2018); we use PyTorch version of the Inception V3 network (Paszke et al., 2017).

**Results on MNIST(s).**   In Fig.(2),(3) we provide results for three MNIST datasets. Both figures depict values averaged over five runs. We report mean values, standard deviations and comparison with the multihead architectures in Supp.(B.1). We evaluate the performance of VAE on the test dataset after each new task is added. The x-axis denotes a total number of tasks seen by the models (and thus, the total number of tasks in the test dataset). The flatter the line is, the less forgetting we observe as a number of tasks increases. We provide numbers for each task separately in Supp.(B.2). Notice that BooVAE can be combined with any other weight-regularization method. We have observed that EWC does not improve performance a lot (see Supp.(B.1)), therefore we report only results for the combination of BooVAE with VCL, which gives the best performance in terms of NLL.

We observe that, based on KL metric, pure BooVAE produces the most diverse samples. We plot several samples from the model after training it on ten tasks in Fig.(4). For MoG, Random Corset and BooVAE, we use the same amount of components equal to 15 for each task. It is worth mentioning that the closer the random corset size is to the size of the training data, the better the performance should be. In the limit case, we can store all the data from the previous tasks, guaranteeing the absence of catastrophic forgetting. Our goal is to reduce the amount of stored information, thus we used a pretty small number of components. In Supp.(B.4) we explore, how large the random corset should be to get the performance comparable to BooVAE.

**Results on CelebA**   We conduct experiments on CelebA dataset for several reasons. Firstly, we want to show that our method works not only on small-scale images, such as MNIST but also on higher-dimensional data. Secondly, since the classes differ only by the hair color in this setting, we may see the advantages of the shared architecture more clearly. In the CelebA dataset there are much fewer faces with grey hair, compared to other colors. Therefore, information from other

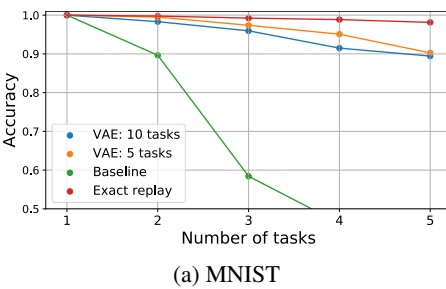 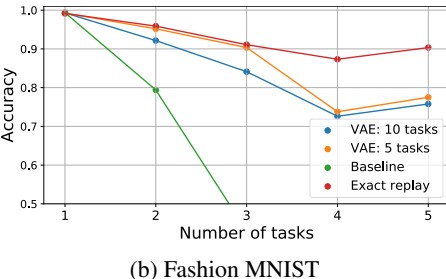

|           |                  |
|:---------:|:----------------:|
| (a) MNIST | (b) Fashion MNIST |

Figure 6: We consider continual learning experiment for MNIST and Fashion MNIST datasets, where we split the dataset in 5 task, containing pairs of disjoint classes, i.e. '0/1', '2/3', etc. We train both BooVAE and classification DNN in continual setting, using VAE for generative replay to avoid catastrophic forgetting in classification.

classes is essential to obtain good results on these images. We compare the FID values as new tasks are added to the models in Table(1). The BooVAE outperforms other approaches, including model with multihead architecture. Notice that multihead fails on the last task, which has much fewer observations, compared to others. This highlights the benefit of the shared architecture, as the Multi-heads approach limits the amount of shared information between tasks. Samples from the different VAEs can be found on Fig.(5). We provide more samples in Supp.(B.3).

## 5.2 Generative Replay for Discriminative Model with continual VAE

**Motivation** Common approach to mitigate catastrophic forgetting in discriminative models is deep generative replay (Shin et al., 2017). The method is based on the recollection of the past knowledge, such as the data of past classes, by generating it from the trained generative model. However, since continual learning for generative models was limited, it was proposed to re-train the generative model from scratch when a new task arrives. Since we propose the method for the continual learning of the generative model, we can avoid this. We can continually train VAE along with the discriminative model.

**Setting** We perform the continual learning experiment using the MNIST and Fashion MNIST datasets. We apply splitMNIST setting when the dataset is split into five binary classification tasks. E.g. the first task containing digits '0' and '1', the second task — digits '2' and '3', and so on. We perform similar splitting into the five tasks for the Fashion MNIST. We train MLP with two hidden layers, LeakyReLU activations and Dropout layers. For each task, we add a new classification head (one fully connected layer) and train for 200 epochs with batch size 500.

**Results** The mean accuracy across all tasks is reported in Fig.(6). We provide two models for comparison. Exact replay setting reuses all the training data from the previous task, thus, giving an upper bound on the generative replay's performance. In the baseline method, we do not use any information about the previous task. This gives us a lower bound on the performance, i.e. if replay buffer is not used. We report two versions of the generative replay with VAE. In the first one, we consider each class as a separate task and train VAE in the continual setting (VAE: 10 tasks) and sample images from the last available model. We use prior components to label the image classes. For example, along with the first classification task on MNIST (with digits '0' and '1') we train VAE firstly on '0's and then on '1's. As a result, we have separate components in the prior distribution for both tasks. Sampling latent vector from these components decoding them gives us the replay buffer for the next classification task. In the second setting, we combine classes like in splitMNIST setting, i.e. each task contains two classes (VAE: 5 tasks). In this setting, samples from the prior have to be classified in order to be used in classification model. We follow the approach from Shin et al. (2017), using classier from the previous step for this purpose. Results that we observe are comparable with the MNIST results in Shin et al. (2017)[3], while our approach avoids full re-training of the generative model.

---

[3]Authors provide only plot (Fig. 2 in their paper), therefore we are not able to report exact numbers for comparison

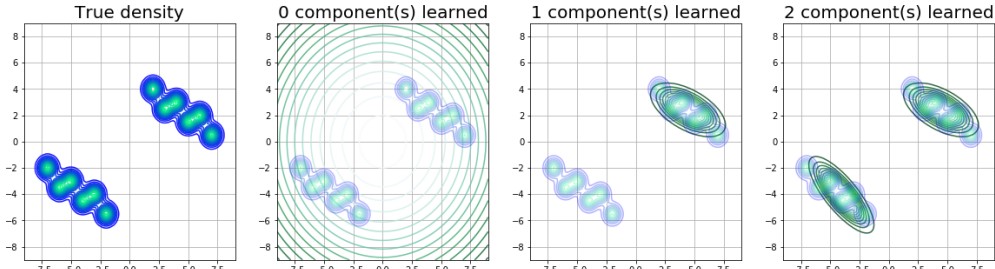

Figure 7: The mixture of 12 Gaussian is approximated by 2-component Gaussian mixture with Eq.(11). This toy experiment illustrates the intuition of our approach, where we hope to efficiently approximate prior $\pi^*(\boldsymbol{z}) = \mathbb{E}_{p_e(\boldsymbol{x})} q_\phi(\boldsymbol{z}|\boldsymbol{x})$ given only empirical version of the target density $\frac{1}{N}\sum_{n=1}^{N} q_\phi(\boldsymbol{z}|\boldsymbol{x}_n)$.

## 6   Conclusion

We propose a novel algorithm for continual learning of VAEs with the static architecture by incorporating the data-driven information about new task into the prior over the latent space. We leverage the specific structure of the VAE model and match new data innovation with the additive aggregated posterior expansion. The boosting-like approach allows us to reduce the number of components in the approximation of the optimal prior distribution without the loss of performance. We empirically validate performance of our algorithm and compare with other approaches. That being said, the proposed algorithm is orthogonal to them and could be easily combined. We would like to finalize with additional comments on performance evaluation and approach intuition.

**Model perfomance**   In the continual learning setting it is important to evaluate the evolution of the metrics for each task while new task arrives. Hence in Sections (B.1) - (B.3) we provide more detailed overview of models performance and report NLL and diversity metric after each additional task is trained. We discuss how the performance changes on the whole test dataset as we keep training in continual learning setting. Moreover, we report and discuss these metrics for each task separately in Section (B.2). Finally, we visually study the samples from the model on each step in Section (B.3).

**Approach intuition**   We provide toy-example visualization in Fig.(8,7) to illustrate equations at Sec.(3).

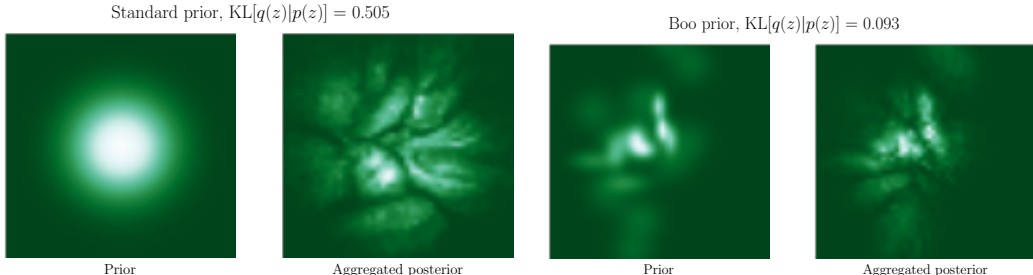

Figure 8: Visualization of the prior density and aggregate posterior for the VAE with 2-d latent space. In left we present results for the VAE with $\mathcal{N}(\boldsymbol{z}|\boldsymbol{0}, I)$ prior and in the right with the proposed Boo prior. By visual inspection and KL-divergence comparison we conclude that the proposed prior matches the aggregated posterior better that a standard normal prior.

## Acknowledgments

Authors are thankful to Jakub Tomczak for feedback and inspiring words, to Kirill Neklyudov and Dmitry Molchanov for important discussions during 2019 and to the reviewers who provided detailed feedback. The work of Evgeny Burnaev was supported by Ministry of Science and Higher Education grant No. 075-10-2021-068.

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
