# Supplementary Material
## BooVAE: Boosting Approach for Continual Learning of VAE

**Appendix Organization**

- Section (A) contains technical details of BooVAE algorithm. In A.1 we provide skipped details related to the algorithm derivation: ELBO decomposition, approximated optimal prior, properties of the optimization problem and functional gradient of the objective. Next, we provide derivations for trainable flow-based prior (A.2). Finally, we discuss step-back for selection of the number of components for each task (A.3).

- Section (B) contains broader details and results for experiments in continual framework.

  - In Sections (B.1) - (B.3) we provide more detailed overview of models performance. In Section (B.1) we report NLL and diversity metric after each additional task is trained. We discuss how the performance changes on the whole test dataset as we keep training in continual learning setting. Moreover, we report and discuss these metrics for each task separately in Section (B.2). Finally, we visually study the samples from the model on each step in Section (B.3).

  - In Section (B.4) we provide additional comparison with random coresets, showing that BooVAE requires much less components to get comparable results in term of NLL and diversity metrics.

  - The implementation details for experiments are in Sec. (B.5), including architecture of neural networks and optimization details. The source code is available at https://github.com/AKuzina/BooVAE.

## A  Details of the BooVAE algorithm derivations

### A.1  Derivations for the optimal prior in continual framework

**ELBO decomposition derivation**    We begin with derivations of the used decomposition in Eq.(3). We start form the ELBO definition (1) and conclude with the desirable result with several re-arrangements.

$$
\mathcal{L}(\theta, \phi, \lambda) \triangleq \mathbb{E}_{\substack{p_e(\boldsymbol{x}) \\ q_\phi(\boldsymbol{z}|\boldsymbol{x})}} \log \frac{p_\theta(\boldsymbol{x}|\boldsymbol{z})\pi(\boldsymbol{z})}{q_\phi(\boldsymbol{z}|\boldsymbol{x})} = \mathbb{E}_{\substack{p_e(\boldsymbol{x}) \\ q_\phi(\boldsymbol{z}|\boldsymbol{x})}} \log p_\theta(\boldsymbol{x}|\boldsymbol{z}) + \mathbb{E}_{\substack{p_e(\boldsymbol{x}) \\ q_\phi(\boldsymbol{z}|\boldsymbol{x})}} \log \frac{\pi(\boldsymbol{z})}{q_\phi(\boldsymbol{z}|\boldsymbol{x})} \frac{\hat{q}(\boldsymbol{z})}{\hat{q}(\boldsymbol{z})} =
$$

$$
= \mathbb{E}_{\substack{p_e(\boldsymbol{x}) \\ q_\phi(\boldsymbol{z}|\boldsymbol{x})}} \log p_\theta(\boldsymbol{x}|\boldsymbol{z}) - \Big( \mathbb{E}_{p_e(\boldsymbol{x})} \mathrm{KL}[q_\phi(\boldsymbol{z}|\boldsymbol{x})|\hat{q}(\boldsymbol{z})] + \underbrace{\mathbb{E}_{\substack{p_e(\boldsymbol{x}) \\ q_\phi(\boldsymbol{z}|\boldsymbol{x})}} \log \frac{\hat{q}(\boldsymbol{z})}{\pi(\boldsymbol{z})}}_{①} \Big).
$$

$$(18)$$

This decomposition holds for any choice of the density $\hat{q}(\boldsymbol{z})$ (under mild conditions). We make the specific choice $\hat{q}(\boldsymbol{z}) = \mathbb{E}_{p_e(\boldsymbol{x})} q_\phi(\boldsymbol{z}|\boldsymbol{x})$ and proceed with the term ①, with celebrated Fubini's theorem:

$$
① = \int d\boldsymbol{x} d\boldsymbol{z} \, p_e(\boldsymbol{x}) q_\phi(\boldsymbol{z}|\boldsymbol{x}) \log \frac{\hat{q}(\boldsymbol{z})}{\pi(\boldsymbol{z})} = \int d\boldsymbol{z} \, \log \frac{\hat{q}(\boldsymbol{z})}{\pi(\boldsymbol{z})} \underbrace{\int d\boldsymbol{x} \, p_e(\boldsymbol{x}) q_\phi(\boldsymbol{z}|\boldsymbol{x})}_{=\mathbb{E}_{p_e(\boldsymbol{x})} q_\phi(\boldsymbol{z}|\boldsymbol{x})} =
$$

$$(19)$$

$$
= \int d\boldsymbol{z} \, \hat{q}(\boldsymbol{z}) \log \frac{\hat{q}(\boldsymbol{z})}{\pi(\boldsymbol{z})} = \mathrm{KL}[\hat{q}(\boldsymbol{z})|\pi(\boldsymbol{z})].
$$

We substitute (19) in (18) and obtain the decomposition (3):

$$
\mathbb{E}_{\substack{p_e(\boldsymbol{x}) \\ q_\phi(\boldsymbol{z}|\boldsymbol{x})}} \log \frac{p_\theta(\boldsymbol{x}|\boldsymbol{z})\pi(\boldsymbol{z})}{q_\phi(\boldsymbol{z}|\boldsymbol{x})} = \mathbb{E}_{p_e(\boldsymbol{x})}[\mathbb{E}_{q_\phi(\boldsymbol{z}|\boldsymbol{x})} \log p_\theta(\boldsymbol{x}|\boldsymbol{z}) - (\mathrm{KL}[q_\phi(\boldsymbol{z}|\boldsymbol{x})|\hat{q}(\boldsymbol{z})] + \mathrm{KL}[\hat{q}(\boldsymbol{z})|\pi(\boldsymbol{z})])].
$$

$$(20)$$

Next we proceed to the case of $p_e(\boldsymbol{x}) = \alpha\, p_e^1(\boldsymbol{x}) + (1-\alpha)\, p_e^2(\boldsymbol{x})$, $\alpha \in (0; 1)$. We start with a noticing:

$$\mathbb{E}_{\substack{p_e(\boldsymbol{x}) \\ q_\phi(\boldsymbol{z}|\boldsymbol{x})}} \log \frac{\pi(\boldsymbol{z})}{q_\phi(\boldsymbol{z}|\boldsymbol{x})} \frac{\hat{q}(\boldsymbol{z})}{\hat{q}(\boldsymbol{z})} = \alpha\, \mathbb{E}_{\substack{p_e^1(\boldsymbol{x}) \\ q_\phi(\boldsymbol{z}|\boldsymbol{x})}} \log \frac{\pi(\boldsymbol{z})}{q_\phi(\boldsymbol{z}|\boldsymbol{x})} \frac{\hat{q}_1(\boldsymbol{z})}{\hat{q}_1(\boldsymbol{z})} + (1-\alpha)\, \mathbb{E}_{\substack{p_e^2(\boldsymbol{x}) \\ q_\phi(\boldsymbol{z}|\boldsymbol{x})}} \log \frac{\pi(\boldsymbol{z})}{q_\phi(\boldsymbol{z}|\boldsymbol{x})} \frac{\hat{q}_2(\boldsymbol{z})}{\hat{q}_2(\boldsymbol{z})}. \tag{21}$$

We select $\hat{q}_i(\boldsymbol{z}) = \mathbb{E}_{p_e^i(\boldsymbol{x})} q_\phi(\boldsymbol{z}|\boldsymbol{x})$, $i = 1, 2$ and with direct using derivations above conclude with the desirable decomposition (5):

$$L(\theta, \phi, \lambda) = \mathbb{E}_{\substack{p_e(\boldsymbol{x}) \\ q_\phi(\boldsymbol{z}|\boldsymbol{x})}} \log p_\theta(\boldsymbol{x}|\boldsymbol{z}) - \sum_{i=1,2} \alpha_i \left( \mathbb{E}_{p_e^i(\boldsymbol{x})} \mathrm{KL}[q_\phi(\boldsymbol{z}|\boldsymbol{x})|\hat{q}_i(\boldsymbol{z})] + \mathrm{KL}[\hat{q}_i(\boldsymbol{z})|\pi(\boldsymbol{z})] \right). \tag{22}$$

**Proof of the approximated optimal prior in (7)**   We consider the modified optimization problem over the probability density space $\mathcal{P}$ (6) with the approximation of the aggregated variational posterior of the first task $\hat{q}_1^a(\boldsymbol{z}) \approx \hat{q}_1(\boldsymbol{z})$:

$$\min_{\pi(\boldsymbol{z}) \in \mathcal{P}} \alpha\, \mathrm{KL}[\hat{q}^a(\boldsymbol{z})|\pi(\boldsymbol{z})] + (1-\alpha)\, \mathrm{KL}[\hat{q}_2(\boldsymbol{z})|\pi(\boldsymbol{z})]. \tag{23}$$

The optimization problem (23) is convex over $\pi(\boldsymbol{z})$ as the sum of convex functions, hence we proceed with FOC of the corresponded Lagrangian with normalization conditions:

$$\frac{\delta}{\delta\pi} \alpha\, \mathrm{KL}[\hat{q}^a(\boldsymbol{z})|\pi(\boldsymbol{z})] + (1-\alpha)\, \mathrm{KL}[\hat{q}_2(\boldsymbol{z})|\pi(\boldsymbol{z})] + \lambda \left( \int d\boldsymbol{z}\, \pi(\boldsymbol{z}) - 1 \right) =$$
$$= -\left( \alpha \frac{\hat{q}^a(\boldsymbol{z})}{\pi(\boldsymbol{z})} + (1-\alpha) \frac{\hat{q}_2(\boldsymbol{z})}{\pi(\boldsymbol{z})} \right) + \lambda = 0. \tag{24}$$

By rearranging terms and normalize the solution, we conclude with the stated result:

$$\pi^{1,2}(\boldsymbol{z}) = \alpha \hat{q}_1^a(\boldsymbol{z}) + (1-\alpha) \hat{q}_2(\boldsymbol{z}). \tag{25}$$

**Proof of the bi-convexity of the optimization problem (8) over $h$ and $\beta$**   We consider the functional:

$$\alpha\, \mathrm{KL}[\hat{q}_1^a(\boldsymbol{z})|(1-\beta)\pi^1(\boldsymbol{z}) + \beta h(\boldsymbol{z})] + (1-\alpha)\, \mathrm{KL}[\hat{q}_2(\boldsymbol{z})|(1-\beta)\pi^1(\boldsymbol{z}) + \beta h(\boldsymbol{z})]). \tag{26}$$

We show bi-convexity over $h$ and $\beta$ for the term $\mathrm{KL}[\hat{q}_1^a(\boldsymbol{z})|(1-\beta)\pi^1(\boldsymbol{z}) + \beta h(\boldsymbol{z})]$ as other is of the same form and sum of convex functions is a convex function. The convexity over $h$ follows from convexity of KL divergence:

$$\mathrm{KL}[\hat{q}_1^a(\boldsymbol{z})|(1-\beta)\pi^1(\boldsymbol{z}) + \beta(\alpha h_1(\boldsymbol{z}) + (1-\alpha)h_2(\boldsymbol{z})] \leq$$
$$\leq (1-\beta)\, \mathrm{KL}[\hat{q}_1^a(\boldsymbol{z})|\pi^1(\boldsymbol{z})] + \beta\, (\alpha\, \mathrm{KL}[\hat{q}_1^a(\boldsymbol{z})|h_1(\boldsymbol{z})] + (1-\alpha)\, \mathrm{KL}[\hat{q}_1^a(\boldsymbol{z})|h_2(\boldsymbol{z})]). \tag{27}$$

We check convexity over $\beta$ by expecting of the second derivative:

$$\nabla_\beta \mathrm{KL}[\hat{q}_1^a(\boldsymbol{z})|(1-\beta)\pi^1(\boldsymbol{z}) + \beta h(\boldsymbol{z})] = -\int d\boldsymbol{z}\, \hat{q}_1^a(\boldsymbol{z}) \frac{h(\boldsymbol{z}) - \pi^1(\boldsymbol{z})}{(1-\beta)\pi^1(\boldsymbol{z}) + \beta h(\boldsymbol{z})},$$
$$\nabla_\beta^2 \mathrm{KL}[\hat{q}_1^a(\boldsymbol{z})|(1-\beta)\pi^1(\boldsymbol{z}) + \beta h(\boldsymbol{z})] = \int d\boldsymbol{z}\, \hat{q}_1^a(\boldsymbol{z}) \left( \frac{h(\boldsymbol{z}) - \pi^1(\boldsymbol{z})}{(1-\beta)\pi^1(\boldsymbol{z}) + \beta h(\boldsymbol{z})} \right)^2 > 0. \tag{28}$$

**Derivation of the functional gradient for the optimization problem (8) over $h$**   The functional of our interest is $\alpha\, \mathrm{KL}[\hat{q}_1^a(\boldsymbol{z})|(1-\beta)\pi^1(\boldsymbol{z}) + \beta h(\boldsymbol{z})] + (1-\alpha)\, \mathrm{KL}[\hat{q}_2(\boldsymbol{z})|(1-\beta)\pi^1(\boldsymbol{z}) + \beta h(\boldsymbol{z})])$. We consider the perturbation of the argument $\pi^1(\boldsymbol{z})$ with $h(\boldsymbol{z})$ as following $(1-\beta)\pi^1(\boldsymbol{z}) + \beta h(\boldsymbol{z})$. We start from the linearization of the first term:

$$\mathrm{KL}[\hat{q}_1^a(\boldsymbol{z})|(1-\beta)\pi^1(\boldsymbol{z}) + \beta h(\boldsymbol{z})] = -\int d\boldsymbol{z}\, \hat{q}_1^a(\boldsymbol{z}) \Big\{ \log \frac{\pi^1(\boldsymbol{z})}{q_1^a(\boldsymbol{z})} + \log \left[ 1 + \beta \left( \frac{h(\boldsymbol{z})}{\pi^1(\boldsymbol{z})} - 1 \right) \right] \Big\} =$$
$$= \{ \log(1+x) = x + o(x), \} = \mathrm{KL}[\hat{q}_1^a(\boldsymbol{z})|\pi^1(\boldsymbol{z})] - \beta \left( \int d\boldsymbol{z}\, h(\boldsymbol{z}) \frac{\hat{q}_1^a(\boldsymbol{z})}{\pi^1(\boldsymbol{z})} - 1 \right) + o(\beta). \tag{29}$$

With application of this result to the second term, we obtain:

$$\alpha\,\mathrm{KL}[\hat{q}_1^a(\boldsymbol{z})|(1-\beta)\pi^1(\boldsymbol{z})+\beta h(\boldsymbol{z})]+(1-\alpha)\,\mathrm{KL}[\hat{q}_2(\boldsymbol{z})|(1-\beta)\pi^1(\boldsymbol{z})+\beta h(\boldsymbol{z})])=$$

$$\alpha\,\mathrm{KL}[\hat{q}_1^a(\boldsymbol{z})|\pi^1(\boldsymbol{z})]+(1-\alpha)\,\mathrm{KL}[\hat{q}_2(\boldsymbol{z})|\pi^1(\boldsymbol{z})]-\beta\underbrace{\left(\int dz\,h(z)\left[\alpha\frac{\hat{q}_1^a(\boldsymbol{z})}{\pi^1(\boldsymbol{z})}+(1-\alpha)\frac{\hat{q}_2(\boldsymbol{z})}{\pi^1(\boldsymbol{z})}\right]-1\right)}_{①}+o(\beta).$$

$$(30)$$

The term ① is the functional gradient. We project direction $h$ to match it at the optimization problem (12).

## A.2 BooVAE for VAE with flow-based prior

Consider the ELBO objective:

$$\mathcal{L}(\theta,\phi)\triangleq\mathbb{E}_{p_e(\boldsymbol{x})q_\phi(\boldsymbol{z}|\boldsymbol{x})}\left(\log p_\theta(\boldsymbol{x}|\boldsymbol{z})\pi(\boldsymbol{z})-\log q_\phi(\boldsymbol{z}|\boldsymbol{x})\right). \tag{31}$$

The simplest choice of the prior $p(\boldsymbol{z})$ is the standard normal distribution. Instead, in order to improve ELBO, one could obtain multi-modal prior distribution $p(\boldsymbol{z})$ by using learnable bijective transformation $f$:

$$\boldsymbol{v}\sim p_0(\boldsymbol{v}),\boldsymbol{z}=f(\boldsymbol{v}). \tag{32}$$

This induce the following density over the prior in $\boldsymbol{z}$-space:

$$\pi(\boldsymbol{z})=\int p(\boldsymbol{v})\delta_{\boldsymbol{z}}(f(\boldsymbol{v}))\,d\boldsymbol{v}=p_0(f^{-1}(\boldsymbol{z}))|J_{\boldsymbol{z}}^{f^{-1}}|. \tag{33}$$

We could obtain optimal prior in z-space by using the following decomposition of the ELBO:

$$\mathbb{E}_{p_e(\boldsymbol{x})}\left[\mathbb{E}_{q_\phi(\boldsymbol{z}|\boldsymbol{x})}\log p_\theta(\boldsymbol{x}\|\boldsymbol{z})-\mathrm{KL}[q_\phi(\boldsymbol{z}|\boldsymbol{x})\|\hat{q}(\boldsymbol{z})]-\mathrm{KL}[\hat{q}(\boldsymbol{z})\|\pi(\boldsymbol{z})]\right], \tag{34}$$

where $\hat{q}(\boldsymbol{z})=\mathbb{E}_{p_e(\boldsymbol{x})}q_\phi(\boldsymbol{z}|\boldsymbol{x})$ is the aggregated variational posterior. As KL-divergence is non-negative, the global maximum over $\pi$ is reached when: $\pi(\boldsymbol{z})=\mathbb{E}_{p_e(\boldsymbol{x})}q_\phi(\boldsymbol{z}|\boldsymbol{x})$. In oder to reach it, we should match:

$$\pi(\boldsymbol{z})=\hat{q}(\boldsymbol{z})\Longrightarrow p_0(f^{-1}(\boldsymbol{z}))|J_{\boldsymbol{z}}^{f^{-1}}|=\mathbb{E}_{p_e(\boldsymbol{x})}q_\phi(\boldsymbol{z}|\boldsymbol{x}). \tag{35}$$

It could be done with tuning base distribution $p_0$ and parameters of the transformation $f$. In order to use BooVAE algorithm, we decouple this updates. The parameters of the transformation $f$ are updated in the Maximization step, together with encoder $q_\phi(\boldsymbol{z}|\boldsymbol{x})$ and decoder $p_\theta(\boldsymbol{x}|\boldsymbol{z})$ and base prior distribution $p_0(\cdot)$ is fixed. So this step is the same as optimization step in VAE training. In the Minorization step, we need to update $p_0(\cdot)$. In order to do this, we came back to the $\boldsymbol{v}$-space.

$$\mathrm{KL}[\pi(\boldsymbol{z})|\hat{q}(\boldsymbol{z})]=\int\pi(\boldsymbol{z})\log\frac{\pi(\boldsymbol{z})}{\hat{q}(\boldsymbol{z})}\,d\boldsymbol{z}=\left\{\begin{smallmatrix}\boldsymbol{z}=f(\boldsymbol{v}),\\d\boldsymbol{z}=|J_{\boldsymbol{v}}^f|d\boldsymbol{v}\end{smallmatrix}\right\}=$$

$$=\int p_0(\boldsymbol{v})|J_{f(\boldsymbol{v})}^{f^{-1}}|\log\frac{p_0(f^{-1}(f(\boldsymbol{v})))|J_{f(\boldsymbol{v})}^{f^{-1}}|}{\hat{q}(f(\boldsymbol{v}))}\,|J_{\boldsymbol{v}}^f|d\boldsymbol{v}= \tag{36}$$

$$\int p_0(\boldsymbol{v})\log\frac{p_0(\boldsymbol{v})}{\hat{q}(f(\boldsymbol{v})|J_{\boldsymbol{v}}^f|}\,d\boldsymbol{v}=\mathrm{KL}[p_0(\boldsymbol{v})|\hat{q}(f(\boldsymbol{v})|J_{\boldsymbol{v}}^f|].$$

We conclude with the same problem of matching aggregation posterior, but in the $\boldsymbol{v}$-space.

## A.3 Step-back for components

On practice, it is not obvious, what is the optimal number of components in the prior. We have experimentally observed, that excessive amount of the prior component can be as harmful as the insufficient number of components. This happens because we learn each component as a pseudoinput to the encoder. Throughout VAE training we update parameters of the encoder by maximizing the ELBO, meaning that components are also unintentionally updated and some of them become irrelevant.

To circumvent this disadvantage without the need to retrain VAE with different number components, we suggest to prune the prior after the maximal number of components is reached. This approach

allows us in theory to select this maximal number to be as large as possible and then remove all the relevant ones during pruning.

Pruning is performed as minimization of the KL-divergence between optimal prior and current approximation with the respect to the weights if the mixture. This optimization procedure is performed in the latent space and only ones during training. Therefore, it almost does not influence the training time. In the experiment setting we report the **maximal** number of components. That is total number of prior components before pruning.

# B   Details of the Experiments and Ablation Study

All the experiments we performed on a single NVIDIA Tesla V100 GPU.

## B.1   Results in continual learning setting

In Tables (2), (3) we provide full results for continual learning experiments, which are illustrated by Fig.(2),(3). We provide negative log-likelihood in Table (2) and KL-divergence used as diversity measure in Table (3). The first column in both tables states how many tasks did the VAE see in total and the value in the table indicates value of negative log-likelihood or diversity metrics on the test set containing current and all the previous tasks.

We add comparison with multi-head architectures. We observe that BooVAE is capable to achieve results comparable to multi-head architecture. We find this to be a good result, because fixed architecture that we use has approximately 6 times less parameters. Moreover, when computing NLL on the test set multi-head architecture requires task tables in order to use proper head, while in case of BooVAE the test evaluation is performed in unsupervised manner. Even though NLL on a test set is a default way of assessing VAEs, during our experiments, we've observed that good NLL scores do not always correspond to diverse samples from the prior in the continual setting. See samples in Fig.(12),(13),(14) for the confirmation of this observation. For example, random coresets are performing well in terms of NLL, but we can see that they are still prone to catastrophic forgetting when it comes to generating samples. Therefore, we were interested in quantitative evaluation of the samples, produced by the VAE on each step. As the diversity score we calculate KL-divergence between desired $P(x)$ and observed $Q(x)$ distribution of generated images over the classes. Desired distribution is multinomial with equal probabilities for each class. To evaluate observed distribution of classes we generate $N = 10^3$ samples from VAE and classify them using neural network, trained on the same dataset as the VAE. Then we calculate the proportion of images from the class $i$ in the generated sample and evalutate KL-divergence:

$$P(x=i)=p_i=\frac{1}{T},\ i\in\{1,\dots,T\}, Q(x=i)=\hat{p}_i=\frac{N_i}{N},\ i\in\{1,\dots,T\},\ \mathrm{KL}[P(x)|Q(x)]=\sum_{i=1}^{T}\frac{1}{T}\log\frac{\frac{1}{T}}{\hat{p}_i}.$$

(37)

If the model generates images from all the classes in equal proportions, the value of the metrics is zero. The large is KL-divergence, the less diversity is there in the samples. We would like to note, that this metric is reflecting the situation that we observe in Figures 12, 13, 14 and confirms that BooVAE is dealing with catastrophic forgetting much better that other methods. Moreover, we observe that even though the combination of BooVAE with VCL is the best in terms of reconstruction error (NLL), it produce more blurred samples and therefore the classification network make more error with is reflected on the KL-divergence.

## B.2   Metrics for each task separately

In Fig.(9),(10) we report test NLL and diversity metrics from the previous section for each task separately. Each subgraph shows performance of the VAE on one specific task (e.g. on '0's, '1's, etc. for MNIST), depending on the total number of tasks seen by the model in continual setting. Each line begins when the class appears in the train set for the first time. We observe, how the performance on this task changes as wee keep updating the model on the new tasks. We expect the line to be as flat as possible. This means, that quality of reconstructions and proportion of samples does not get worse when new tasks arrive. In Fig.(10) we show each term in the KL-divergence as a characteristic of

Table 2: NLL on a test set, averaged over 5 runs with standard deviation in the brackets. Each model was trained on 10 tasks in a continual setting, in multi-head models new encoder and extra decoder layer was added for every new task. We use ***bold italics*** to denote the best result among all the models and **bold** to denote best among the models with one encoder-decoder pair for all the tasks.

| # Tasks | Standard | EWC | VCL | Coreset | MoG | Boo | Boo + VCL | Multihead | Multihead + EWC |
|---|---|---|---|---|---|---|---|---|---|
| | | | | | MNIST | | | | |
| 2 | 343.5 (26) | 258.3 (12) | 84.8 (0.4) | 108.6 (2.2) | 98.1 (3.2) | 96.1 (0.9) | ***74.9 (0.3)*** | 399.2 (6.8) | 119.6 (18.5) |
| 3 | 122.0 (2.3) | 118.9 (1.5) | 107.6 (0.6) | 106.9 (0.2) | 106.9 (2.7) | 101.3 (0.4) | ***95.2 (0.7)*** | 203.9 (1.5) | 114.2 (6.9) |
| 4 | 146.1 (0.3) | 138.5 (1.5) | 118.3 (0.5) | 121.8 (1.3) | 123.7 (0.8) | 111.9 (0.1) | ***104.1 (0.9)*** | 217.6 (3.5) | 115.3 (7.0) |
| 5 | 197.0 (5.7) | 187.4 (2.2) | 129.9 (1.2) | 138.1 (1.9) | 141.2 (3.5) | 121.0 (2.3) | ***109.1 (0.8)*** | 281.5 (2.6) | 115.9 (8.0) |
| 6 | 164.3 (3.8) | 158.8 (2.8) | 126.1 (0.8) | 135.8 (1.1) | 144.5 (3.4) | 120.8 (0.2) | ***113.5 (0.9)*** | 215.0 (2.3) | ***113.5 (5.8)*** |
| 7 | 205.2 (5.6) | 184.2 (3.8) | 128.0 (0.8) | 144.2 (1.7) | 161.9 (5.8) | 122.9 (0.3) | ***113.7 (0.8)*** | 247.6 (4.2) | ***113.7 (6.1)*** |
| 8 | 213.2 (9.2) | 187.0 (2.5) | 125.6 (0.6) | 141.4 (2.5) | 168.7 (7.8) | 122.4 (0.8) | ***111.2 (0.4)*** | 301.0 (5.8) | ***112.2 (6.6)*** |
| 9 | 171.0 (3.6) | 157.7 (3.7) | 127.4 (0.5) | 137.5 (2.8) | 163.3 (2.4) | 124.5 (2.7) | ***114.5 (0.9)*** | 210.6 (5.0) | ***112.3 (5.9)*** |
| 10 | 186.8 (2.3) | 169.3 (3.2) | 125.7 (0.6) | 137.0 (3.5) | 180.1 (1.3) | 124.5 (1.9) | ***112.7 (1.1)*** | 256.5 (6.3) | ***111.3 (5.6)*** |
| | | | | | notMNIST | | | | |
| 2 | 329.5 (24) | 298.9 (9.0) | 221.6 (3.1) | 241.8 (2.4) | 251.1 (4.7) | 238.6 (0.7) | **210.4 (0.9)** | 497.9 (21) | ***188.1 (1.7)*** |
| 3 | 412.3 (23) | 346.5 (7.1) | 234.1 (5.5) | 243.5 (4.6) | 288.5 (11) | 245.6 (3.1) | **206.3 (1.7)** | 702.6 (58) | ***179.8 (1.8)*** |
| 4 | 299.2 (6.2) | 290.3 (8.2) | 253.4 (5.4) | 227.5 (1.4) | 271.2 (6.3) | 238.8 (3.6) | **215.7 (2.3)** | 662.5 (38) | ***175.2 (3.5)*** |
| 5 | 321.4 (9.6) | 290.1 (4.7) | 238.5 (3.8) | 235.0 (4.9) | 274.2 (4.6) | 251.1 (7.1) | **212.7 (1.1)** | 644.0 (33) | ***174.2 (2.3)*** |
| 6 | 329.1 (15) | 316.8 (11) | 248.3 (4.3) | 228.1 (3.5) | 301.7 (18) | 247.5 (11) | **214.6 (0.9)** | 803.7 (72) | ***170.2 (2.0)*** |
| 7 | 310.9 (20) | 294.3 (3.6) | 241.3 (3.3) | 231.2 (3.1) | 288.8 (7.1) | 255.8 (5.9) | **216.4 (0.9)** | 724.6 (39) | ***170.8 (2.1)*** |
| 8 | 338.6 (35) | 316.4 (14) | 254.5 (7.1) | 230.6 (2.0) | 320.4 (15) | 263.1 (4.2) | **224.2 (1.6)** | 757.9 (47) | ***169.1 (2.0)*** |
| 9 | 339.1 (12) | 314.1 (3.8) | 234.2 (3.1) | 229.0 (2.6) | 317.8 (13) | 252.7 (5.3) | **214.8 (1.3)** | 905.3 (56) | ***161.5 (1.6)*** |
| 10 | 402.4 (13) | 374.4 (12) | 242.2 (3.3) | 233.7 (2.3) | 358.6 (14) | 261.4 (8.7) | **214.3 (1.0)** | 951.2 (47) | ***158.7 (2.2)*** |
| | | | | | fashionMNIST | | | | |
| 2 | 259.1 (4.9) | 270.9 (9.7) | 223.5 (0.6) | 230.3 (1.3) | 256.4 (13) | 224.5 (0.6) | **218.8 (0.6)** | 383.3 (15) | ***231.0 (11.4)*** |
| 3 | 290.9 (3.8) | 284.1 (6.8) | 253.8 (0.4) | 257.8 (0.6) | 266.1 (2.7) | 252.9 (0.6) | ***249.7 (0.4)*** | 327.1 (9.6) | ***250.8 (4.2)*** |
| 4 | 273.5 (2.4) | 269.5 (1.8) | 250.9 (0.6) | 254.1 (1.5) | 273.4 (3.8) | 244.7 (0.4) | ***242.1 (0.6)*** | 336.1 (9.5) | ***240.8 (4.0)*** |
| 5 | 272.0 (1.8) | 268.1 (1.0) | 255.9 (0.4) | 255.2 (0.9) | 266.3 (3.2) | 250.8 (0.8) | ***248.6 (0.2)*** | 333.6 (3.7) | ***247.7 (3.5)*** |
| 6 | 507.8 (46) | 447.4 (20) | 262.7 (2.4) | 258.1 (0.5) | 329.3 (14) | 245.2 (3.2) | ***243.5 (1.7)*** | 675.8 (14.1) | ***242.8 (5.6)*** |
| 7 | 276.5 (3.2) | 271.0 (2.8) | 257.2 (0.5) | 255.7 (0.7) | 266.3 (4.7) | 250.0 (1.1) | ***248.5 (0.3)*** | 382.7 (9.4) | ***247.7 (3.2)*** |
| 8 | 1468 (585) | 540.8 (25) | 254.7 (1.6) | 258.0 (0.3) | 314.8 (18) | 243.5 (1.4) | ***240.5 (0.8)*** | 606.2 (35.7) | ***239.7 (3.0)*** |
| 9 | 310.1 (19) | 284.1 (1.4) | 261.9 (0.9) | 260.8 (0.8) | 285.1 (3.0) | 249.3 (1.1) | ***249.1 (1.1)*** | 502.7 (30.3) | ***248.9 (4.5)*** |
| 10 | 799.9 (273) | 399.2 (27) | 268.6 (1.5) | 263.0 (1.6) | 330.9 (8.4) | 250.6 (4.6) | ***248.7 (0.5)*** | 569.1 (28.4) | ***247.3 (4.7)*** |

the per task diversity, which is equal to the $\frac{1}{K}\left(\log\frac{1}{K} - \log\hat{p}_i\right)$, $i \in \{1, \cdots, K\}$. Ideally, this value should be 0 everywhere. If it is positive, then there is not enough images from a given class in the sample. And vice versa, in case of negative value, there are too many samples from a given class. As we can see from the plots, BooVAE is extremely close to the desired behaviour. If the method suffer from catastrophic forgetting, it produces too many samples when the task is seen for the first time (green zone on the graph) and not enough samples later on (red area), since it forgets how to produce this samples.

Table 3: Diversity results, averaged over 5 runs with standard deviation in the brackets. The lower is better, we use **bold** to denote the best model with one encoder-decoder pair.

| # Tasks | Standard | EWC | VCL | Coreset | MoG | Boo | Boo + VCL |
|---|---|---|---|---|---|---|---|
| | | | | MNIST | | | |
| 2 | 0.69 (0.00) | 0.69 (0.00) | 0.10 (0.02) | 0.69 (0.00) | **0.00 (0.00)** | **0.00 (0.00)** | 0.21 (0.03) |
| 3 | 1.04 (0.00) | 1.01 (0.01) | 0.23 (0.02) | 1.00 (0.01) | **0.00 (0.00)** | **0.00 (0.00)** | 0.29 (0.02) |
| 4 | 1.33 (0.01) | 1.32 (0.02) | 0.24 (0.02) | 1.29 (0.02) | **0.00 (0.00)** | **0.00 (0.00)** | 0.34 (0.02) |
| 5 | 1.57 (0.01) | 1.58 (0.01) | 0.12 (0.02) | 1.54 (0.01) | 0.24 (0.00) | **0.00 (0.00)** | 0.30 (0.02) |
| 6 | 1.68 (0.03) | 1.67 (0.02) | 0.57 (0.04) | 1.58 (0.03) | 0.51 (0.01) | **0.00 (0.00)** | 0.64 (0.01) |
| 7 | 1.92 (0.01) | 1.88 (0.02) | 0.65 (0.04) | 1.82 (0.01) | 0.67 (0.14) | **0.00 (0.00)** | 0.58 (0.02) |
| 8 | 2.02 (0.02) | 2.01 (0.01) | 0.63 (0.04) | 1.91 (0.04) | 0.88 (0.15) | **0.03 (0.05)** | 0.59 (0.02) |
| 9 | 2.09 (0.02) | 2.07 (0.02) | 0.67 (0.02) | 1.90 (0.04) | 1.07 (0.14) | **0.08 (0.11)** | 0.63 (0.02) |
| 10 | 2.26 (0.01) | 2.22 (0.01) | 0.78 (0.02) | 2.09 (0.01) | 1.23 (0.14) | **0.11 (0.15)** | 0.74 (0.01) |
| | | | | notMNIST | | | |
| 2 | 0.62 (0.02) | 0.62 (0.02) | **0.00 (0.00)** | 0.57 (0.05) | 0.22 (0.08) | 0.01 (0.01) | 0.10 (0.03) |
| 3 | 1.04 (0.01) | 1.02 (0.01) | **0.02 (0.01)** | 0.93 (0.03) | 0.53 (0.16) | **0.01 (0.00)** | 0.31 (0.05) |
| 4 | 1.11 (0.09) | 1.12 (0.03) | 0.06 (0.01) | 0.92 (0.05) | 0.72 (0.18) | **0.02 (0.02)** | 0.51 (0.07) |
| 5 | 1.35 (0.04) | 1.32 (0.03) | 0.04 (0.01) | 0.94 (0.04) | 0.91 (0.19) | **0.01 (0.01)** | 0.44 (0.06) |
| 6 | 1.49 (0.08) | 1.52 (0.03) | 0.10 (0.02) | 1.07 (0.06) | 1.01 (0.19) | **0.05 (0.03)** | 0.40 (0.06) |
| 7 | 1.52 (0.12) | 1.51 (0.04) | 0.14 (0.03) | 1.00 (0.05) | 1.22 (0.24) | **0.05 (0.04)** | 0.40 (0.06) |
| 8 | 1.46 (0.14) | 1.53 (0.06) | 0.35 (0.07) | 0.76 (0.03) | 1.40 (0.21) | **0.07 (0.03)** | 0.44 (0.07) |
| 9 | 1.10 (0.39) | 1.09 (0.09) | 0.23 (0.03) | 0.37 (0.08) | 1.49 (0.22) | **0.14 (0.10)** | 0.62 (0.06) |
| 10 | 1.95 (0.06) | 1.94 (0.05) | 0.35 (0.06) | 1.22 (0.03) | 1.59 (0.21) | **0.20 (0.15)** | 0.60 (0.05) |
| | | | | fashionMNIST | | | |
| 2 | 0.68 (0.00) | 0.67 (0.01) | 0.01 (0.00) | 0.65 (0.01) | 0.03 (0.03) | **0.00 (0.00)** | **0.00 (0.00)** |
| 3 | 1.02 (0.00) | 1.00 (0.02) | 0.21 (0.02) | 0.89 (0.03) | 0.19 (0.07) | **0.00 (0.00)** | **0.00 (0.00)** |
| 4 | 1.28 (0.01) | 1.26 (0.02) | 0.55 (0.01) | 1.15 (0.03) | 0.34 (0.11) | **0.00 (0.00)** | **0.00 (0.00)** |
| 5 | 1.34 (0.01) | 1.30 (0.02) | 0.30 (0.01) | 1.10 (0.04) | 0.37 (0.09) | **0.00 (0.00)** | 0.01 (0.00) |
| 6 | 1.78 (0.00) | 1.77 (0.01) | 0.31 (0.02) | 1.68 (0.02) | 0.53 (0.14) | **0.00 (0.00)** | 0.01 (0.00) |
| 7 | 1.31 (0.03) | 1.31 (0.05) | 0.61 (0.03) | 1.11 (0.04) | 0.63 (0.12) | **0.01 (0.01)** | 0.02 (0.01) |
| 8 | 2.08 (0.00) | 2.08 (0.00) | 0.36 (0.02) | 1.96 (0.02) | 0.67 (0.12) | **0.00 (0.00)** | 0.03 (0.01) |
| 9 | 2.07 (0.01) | 2.10 (0.00) | 0.86 (0.04) | 1.96 (0.02) | 0.77 (0.11) | **0.01 (0.01)** | 0.02 (0.01) |
| 10 | 2.24 (0.01) | 2.24 (0.01) | 0.63 (0.03) | 2.03 (0.03) | 0.85 (0.10) | **0.02 (0.02)** | 0.04 (0.01) |

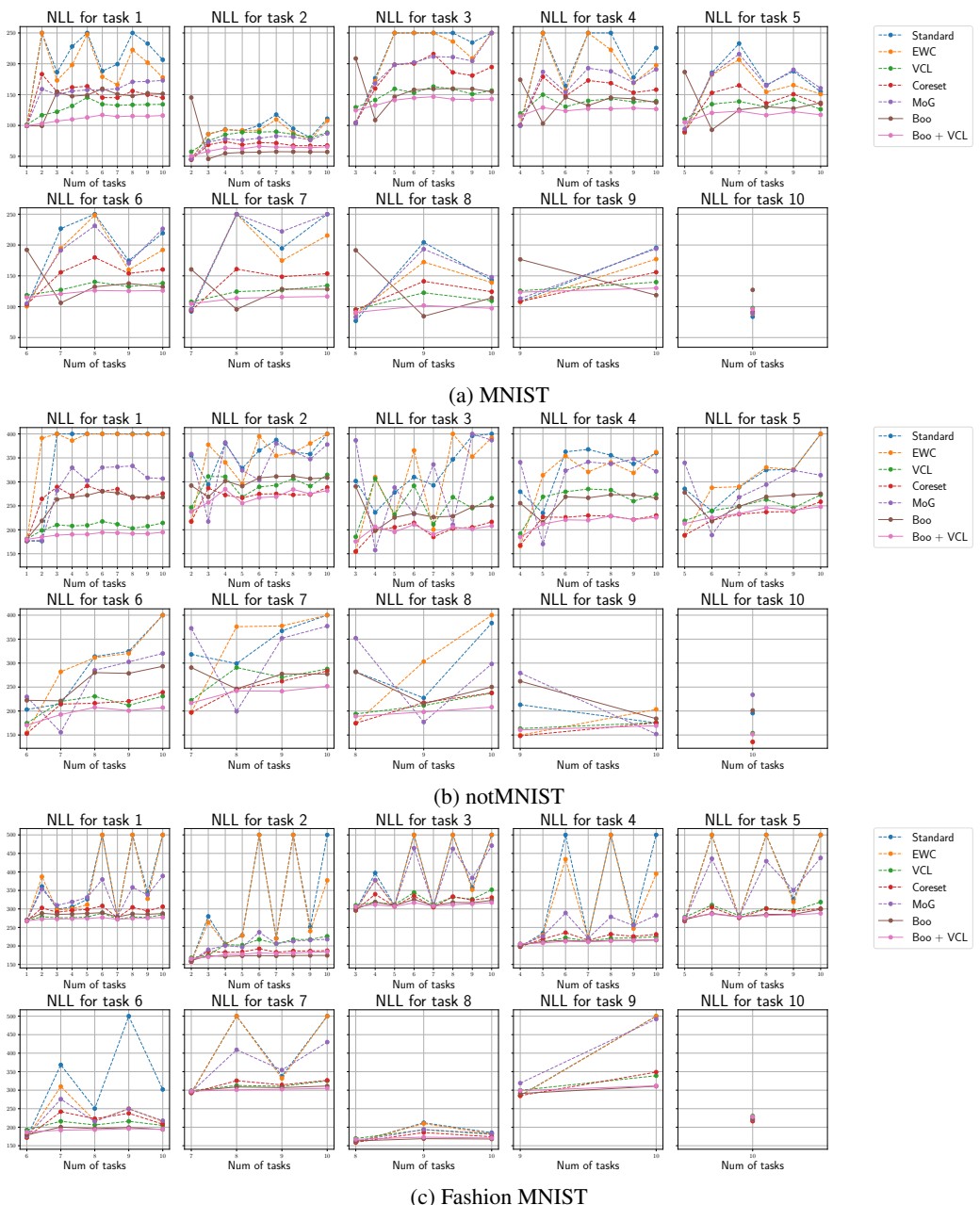

Figure 9: NLL on the test dataset for each task separately averaged over 5 runs. The lower is better.

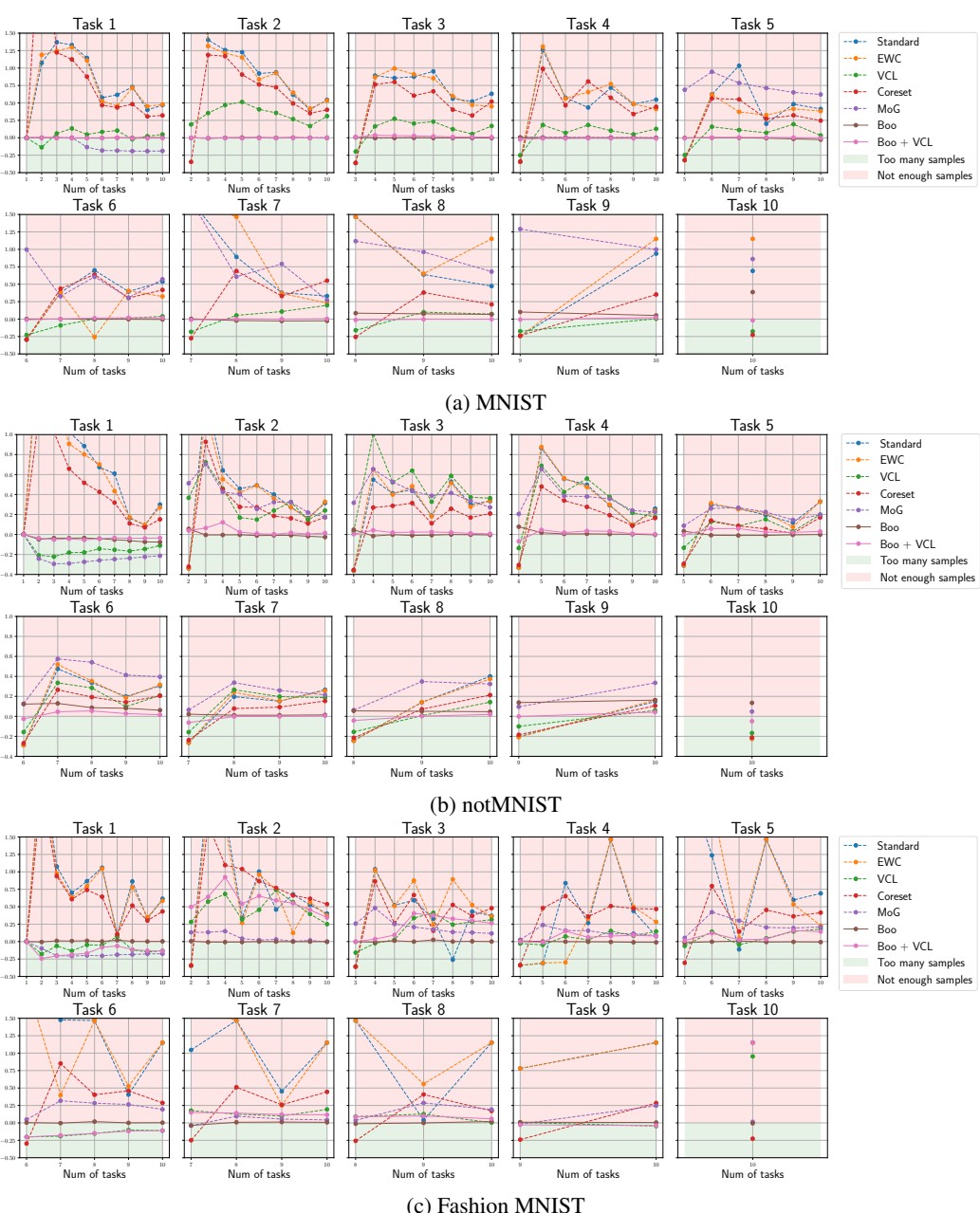

Figure 10: Diversity score for each computed for each class separately. Each subgraph shows values $\frac{1}{K}\left(\log \frac{1}{K} - \log \hat{p_i}\right)$, $i \in \{1, \cdots, K\}$ for the task $k$. That is one term from the sum (KL-divergence), which shows difference between ideal and generated proportion of images for a specific class. On the $x$ axis there is total number of tasks seen by the model. Negative value means that model generates too many images of the class, positive — not enough images from the given class.

## B.3 Examples of generated samples

Figures (11),(12),(13),(14) contain samples from different VAE. Each row-block corresponds to the total number of tasks seen by the model, while each columns corresponds to a different model. We can clearly see, that BooVAE generates diverse samples after training on all the tasks (last row). We also observe that addition of the regularization-based approaches makes samples from the models more blurred. We assume that this is exactly reflected in the worse diversity metrics for BooVAE with VCL. Even though the samples are still diverse, they are are sometimes of lower quality and classification network makes more mistakes.

These samples provide qualitative proof of the generation diversity metrics, shown in Figure (2) and in Tables (3),(1).

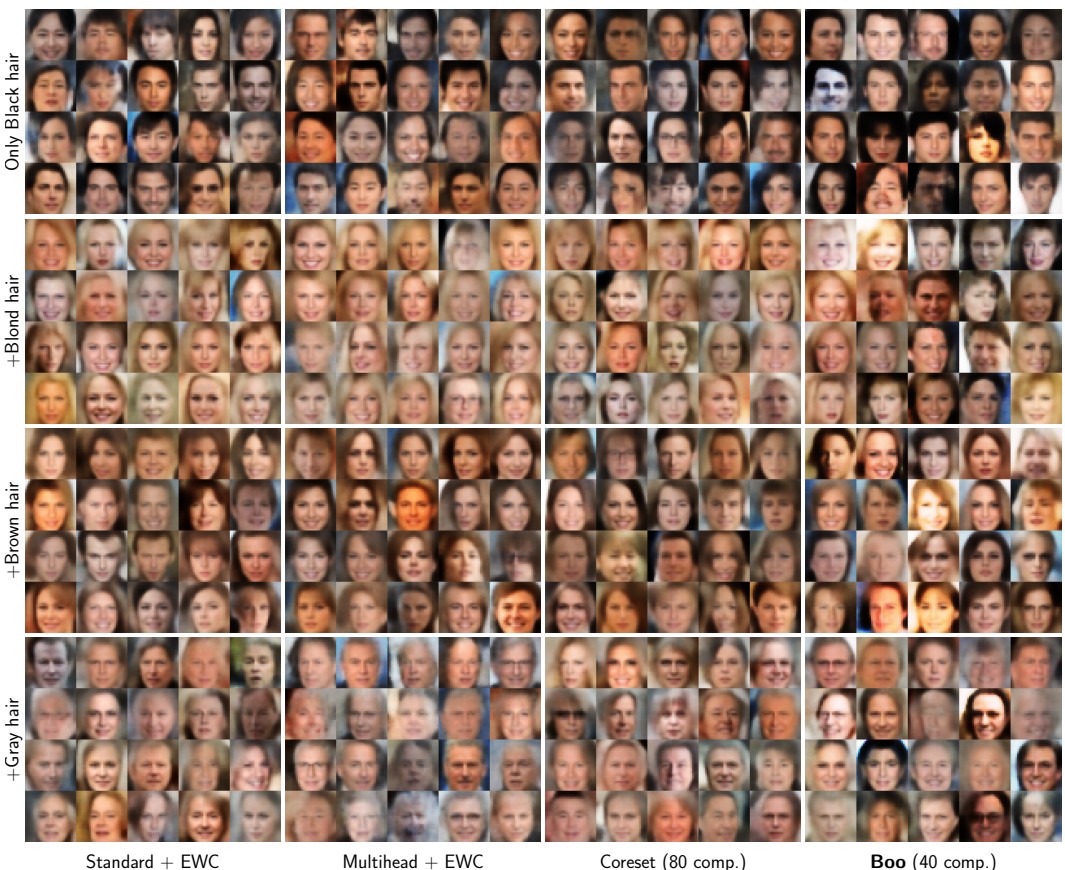

Figure 11: Samples from prior after training on different number of tasks in the continual setting: CelebA dataset. Each row-wise block from the top to the bottom corresponds to the cumulative increasing number of tasks. Each column-wise block corresponds to the particular model.

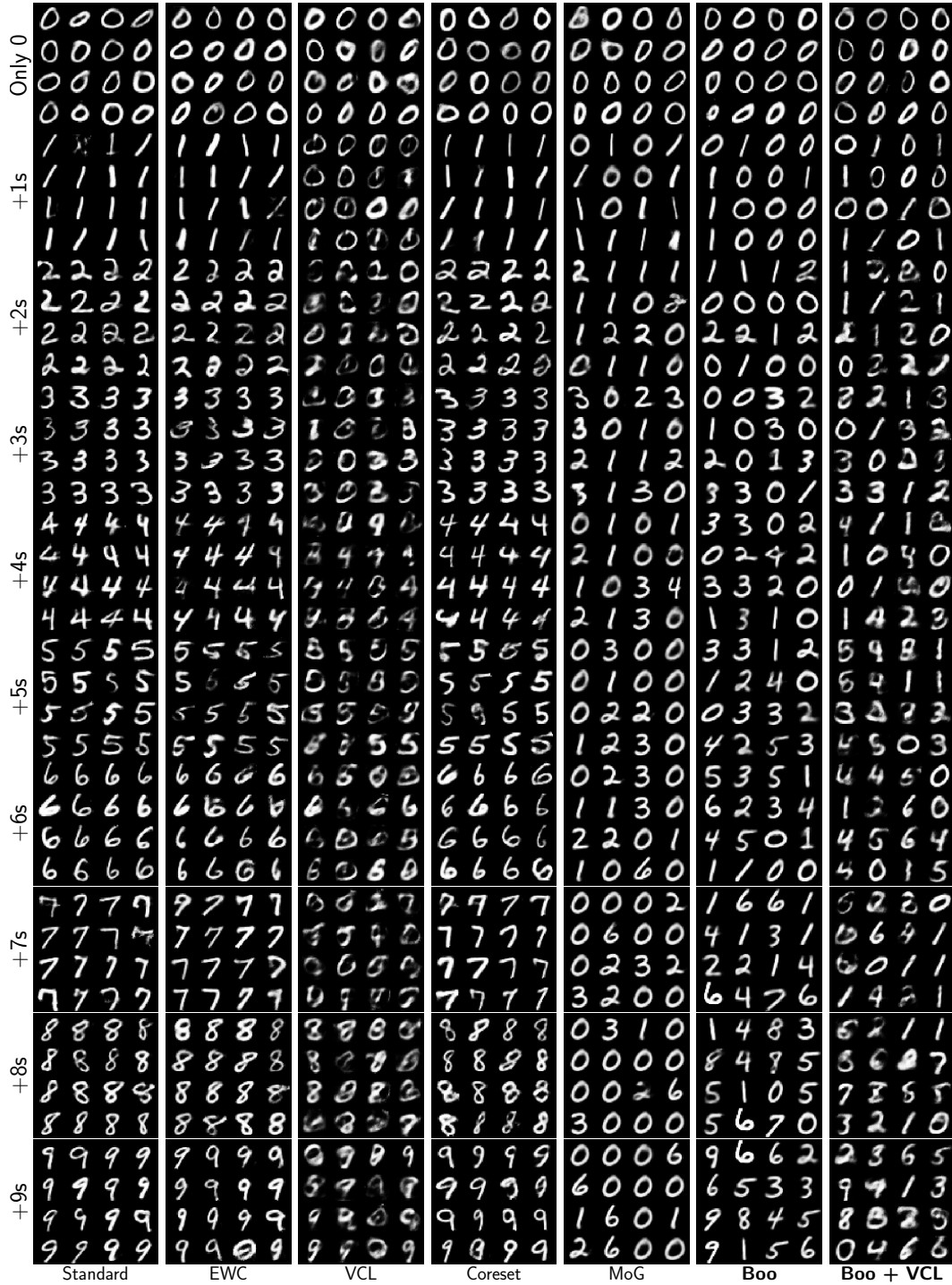

Figure 12: Samples from prior after training on different number of tasks in the continual setting: MNIST dataset. Each row-wise block from the top to the bottom corresponds to the cumulative increasing number of tasks. Each column corresponds to the particular model.

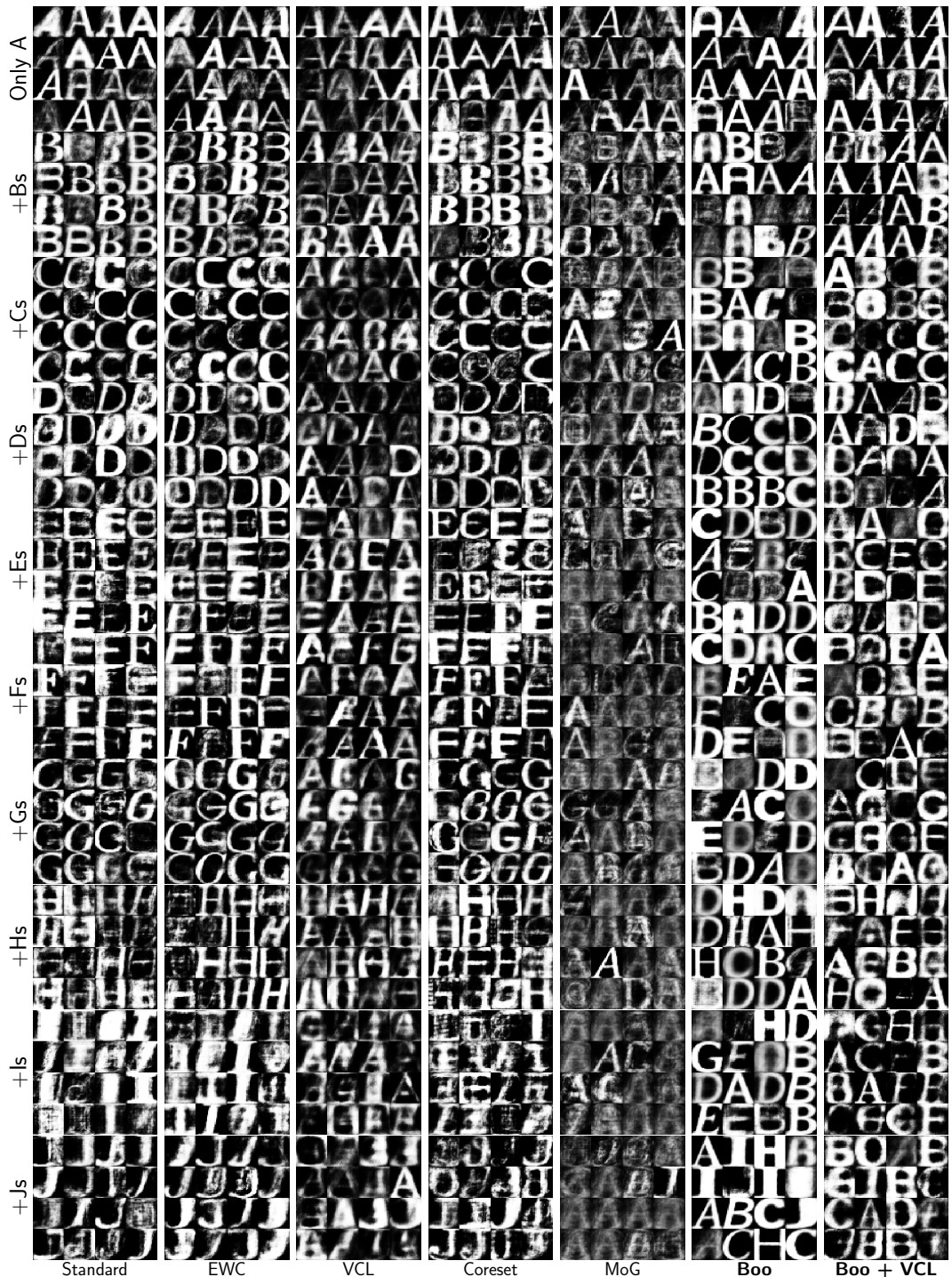

Figure 13: Samples from prior after training on different number of tasks in the continual setting: notMNIST dataset. Each row-wise block from the top to the bottom corresponds to the cumulative increasing number of tasks. Each column-wise block corresponds to the particular model.

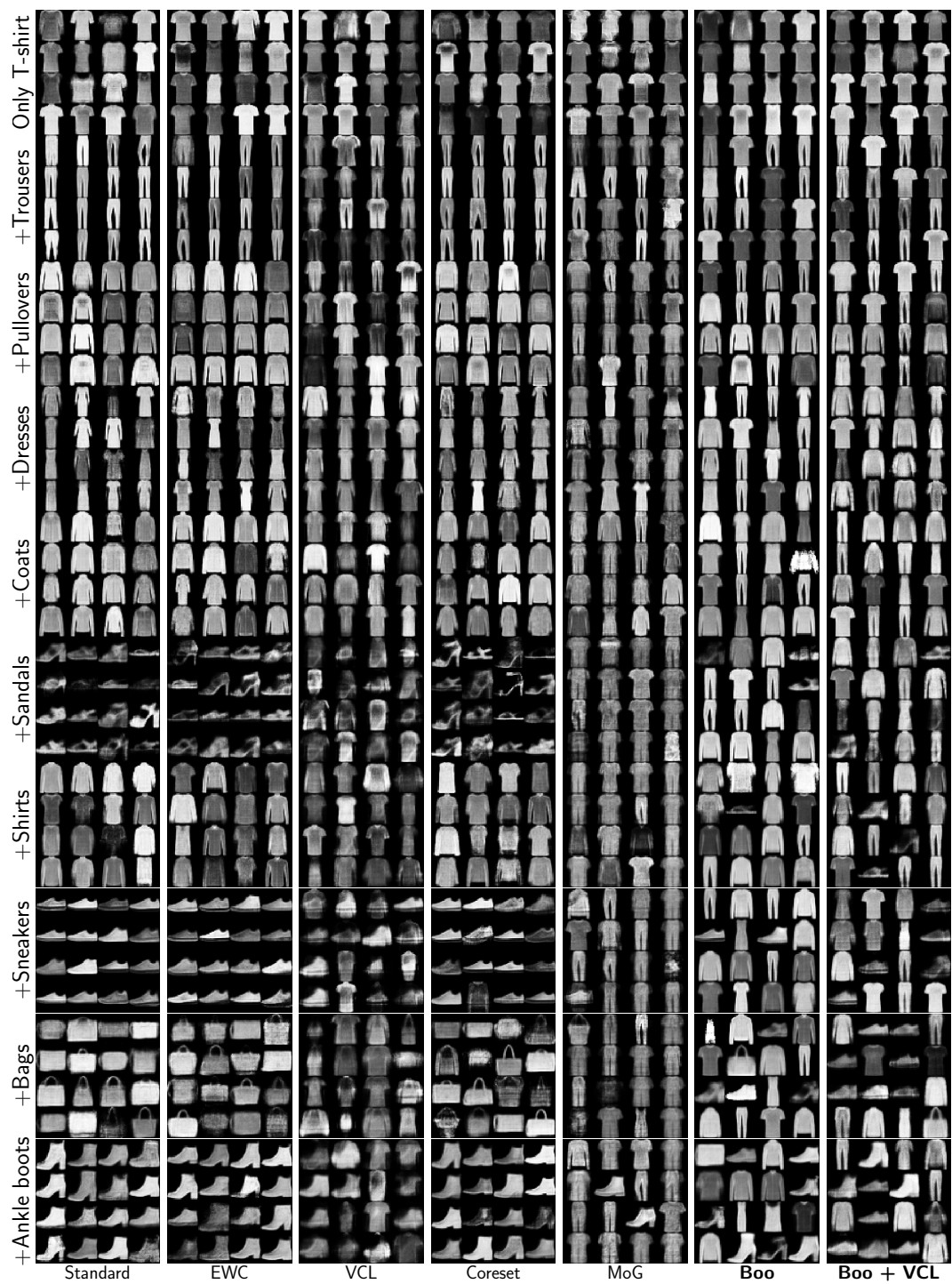

Figure 14: Samples from prior after training on different number of tasks in the continual setting: Fashion MNIST dataset. Each row-wise block from the top to the bottom corresponds to the cumulative increasing number of tasks. Each column-wise block corresponds to the particular model.

## B.4 Random Coreset Size

In all the experiments we use the size of the random coreset equal to the maximal number of components in BooVAE, which is 15 for all the MNIST dataset and 40 for CelebA. Since random coreset basically means that we store a subset of the training data from the previous tasks, it is always possible to find such size of the random coreset, that there is no catastrophic forgetting at all. In this section we show, how large the random coreset should be to achieve results comparable to BooVAE with 15 components.

We observe that on MNIST dataset only random coreset of size 500 per task results in better results in terms of both NLL and KL divergence. Lower size of the random coreset does not produce samples that are as diverse as samples from BooVAE. For Fashion MNIST we observe that situation with NLL is similar, but even 500 samples is not enough to get diverse enough samples from the model.

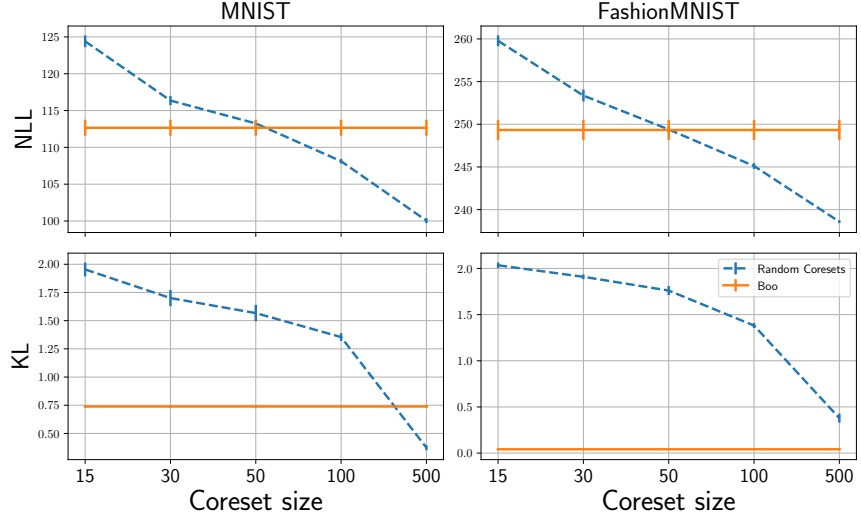

Figure 15: Negative Loglikelohood (top row) and KL, which assesses diversity of generated samples (bottom row) for MNIST and Fashion MNIST dataset for different sizes of Random Corsets after training on 10 tasks.

## B.5 Architecture and Optimization details

**Optimization** We use validation dataset to select hyperparameters for a simple VAE (with a standard Normal prior, trained on the whole training dataset) for each dataset. After that we fix these hyperparemeters for all the methods used in the experiments in the continual learning setting.

For MNIST and FashionMNIST we randomly remove 10'000 images from the train dataset for validation. For notMNIST we use small version of the dataset (19k images in total) and remove 10% as a test data and 10% more as validation. For CelebA dataset we use split provided by the dataset authors.

We use Adam to perform the optimization for all the datasets with LR scheduler, that reduce learning rate by the `factor` when the loss is not decreasing for `patience` number of epochs. In the Table 4 we provide all the parameters of the optimization procedure. For CelebA dataset we use $\beta-$VAE with $\beta$ annealing. The value of $\beta$ is gradually increasing from 0 to 2 during the first 10 epochs.

**MNIST architecture** We've used MLP with 3 linear layers and LeakyReLU activations both for the encoder and decoder in all the cases. Detailed architectures are presented in Table 5.

**CelebA architecture** For CelebA dataset we've used convolutional NN. We use 2D convolutions with kernel size $5 \times 5$, Batch normalization and ReLU activations for encoder and symmetric architecture but with transposed convolutions for the decoder. See all the details in Table 6.

Table 4: Optimization parameters used in the experiments.

| Parameter | MNIST | notMNIST | FashionMNIST | CelebA |
|---|---|---|---|---|
| Batch-size | 250 | 250 | 500 | 512 |
| Initial Learning rate | 5e–4 | 5e–4 | 5e–4 | 1e–3 |
| LR scheduler patience | 30 | 30 | 30 | 9 |
| LR scheduler factor | 0.5 | 0.5 | 0.5 | 0.5 |
| Max epochs | 500 | 500 | 1000 | 300 |
| Early stopping | 50 | 50 | 50 | 15 |
| (# ep. without improvement) | | | | |
| Latent dimension | 40 | 40 | 40 | 128 |
| Beta annealing | No | No | No | From 0 to 2 during first 10 epochs |
| # components per task (BooVAE) | 15 | 15 | 15 | 40 |
| Regularization weight (BooVAE, eq. (17)) | 1 | 1 | 1 | 1 |

Table 5: MLP architectures for MNIST, notMNIST and FashionMNIST datasets.

| MNIST | notMNIST | FashionMNIST |
|---|---|---|
| | Encoder | |
| `Linear(784 -> 300)` | `Linear(784 -> 1024)` | `Linear(784 -> 1024)` |
| `LeakyReLU()` | `LeakyReLU` | `LeakyReLU` |
| `Linear(300 -> 300)` | `Linear(1024 -> 1024)` | `Linear(1024 -> 1024)` |
| `LeakyReLU()` | `LeakyReLU` | `LeakyReLU` |
| $\mu_z \leftarrow$ `Linear(300 -> 40)` | $\mu_z \leftarrow$ `Linear(1024 -> 40)` | $\mu_z \leftarrow$ `Linear(1024 -> 40)` |
| $\log \sigma_z^2 \leftarrow$ `Linear(300 -> 40)` | $\log \sigma_z^2 \leftarrow$ `Linear(1024 -> 40)` | $\log \sigma_z^2 \leftarrow$ `Linear(1024 -> 40)` |
| | Decoder | |
| `Linear(40 -> 300)` | `Linear(40 -> 1024)` | `Linear(40 -> 1024)` |
| `LeakyReLU()` | `LeakyReLU` | `LeakyReLU` |
| `Linear(300 -> 300)` | `Linear(1024 -> 1024)` | `Linear(1024 -> 1024)` |
| `LeakyReLU()` | `LeakyReLU` | `LeakyReLU` |
| `Linear(300 -> 784)` | `Linear(1024 -> 784)` | `Linear(1024 -> 784)` |
| $\mu_x \leftarrow$ `Sigmoid()` | $\mu_x \leftarrow$ `Sigmoid()` | $\mu_x \leftarrow$ `Sigmoid()` |

Table 6: Convolutional architecture for CelebA dataset.

| Encoder | Decoder |
|---|---|
| `Conv(5x5, 3 -> 32)` | `Linear(128 -> 4096)` |
| `BatchNorm()` | `ReLU()` |
| `ReLU()` | `ConvTranspose(5x5, 256 -> 128)` |
| `Conv(5x5, 32 -> 64)` | `BatchNorm()` |
| `BatchNorm()` | `ReLU()` |
| `ReLU()` | `ConvTranspose(5x5, 128 -> 64)` |
| `Conv(5x5, 64 -> 128)` | `BatchNorm()` |
| `BatchNorm()` | `ReLU()` |
| `ReLU()` | `ConvTranspose(5x5, 64 -> 32)` |
| `Conv(5x5, 128 -> 256)` | `BatchNorm()` |
| `BatchNorm()` | `ReLU()` |
| `ReLU()` | `Conv(1x1, 32 -> 3)` |
| $\mu_z \leftarrow$ `Linear(256 -> 128)` | $\mu_x \leftarrow$ `Softsign()` |
| $\log \sigma_z^2 \leftarrow$ `Linear(256 -> 128)` | |