# OpenReview forum: "BooVAE: Boosting Approach for Continual Learning of VAE"
_NeurIPS.cc/2021/Conference — NeurIPS 2021 Poster_

### Official Review · Reviewer_1sJx · 2021-06-25

**Rating:** 6
**Confidence:** 4

**Summary:**

This work introduces the greedy-boosting approach to approximate the posterior for tasks in a continual learning setup for VAE. Its main contributions is the proposed relationship between the approximation of optimal prior, aggregated prior and the learning task at hand, which leads to the introduction of the BooVAE method.


**Limitations And Societal Impact:**

This part is completely missing (together with the conclusions section).

**Main Review:**

The paper is an interesting extension of [VAE with a VampPrior, Jakub M. Tomczak, Max Welling] with the application to continual learning. The idea of parametrizing each task's prior with own variables is novel and original, although a bit incremental. The concept of leveraging pseudo-inputs to approximate an optimal prior distribution per task looks fresh to me. I like the relationship between the proposed method and AdaBoost in terms of Beta coefficient computation. I believe the paper is over-formalized and simple intuitions behind the method should be introduced to increase its impact.

Below I outline the major comments:

Motivation for VAE selection missing
In terms of the significance, I miss the motivation of focusing on VAE in terms of solving for continual learning. The proposed approach relates only to a single generative model, and it is not straightforward to me how to generalize this method to other generative models that can be used in generative replay, e.g. GANs.

Dataset selection
I would recommend extending the dataset pool with Omniglot and validating the performance of the method when the dataset changes (for instance train on MNIST, validate on FashionMNIST, etc. This becomes a standard of continual learning evaluation pipeline and gives more visibility in the algorithm performance.

Requirements formulation
One of the requirements formulated is related to a generative replay and it claims that the generative replay methods need to assess (there's a missing 's') samples quality and task-balance, which leads to a computational overload. This requirement ends with the claim that the proposed approach should avoid generative replay. If I understand correctly this concerns the learning of a VAE (not a general continual learning setup), as in the evaluation section the authors use generative replay as a baseline. This should be clarified.

ELBO Maximization Step
The authors introduce the function that allows to train VAE parameters, but it remains unclear if the ELBO is correctly defined here. The authors claim that they introduce a symmetric KLD, yet I'm unsure if it doesn't conflict with the lower bound.

Minorization-Maximization algorithm
The authors use this term as if it is a well known method, while it seems to be the one introduced in this work. Please clarify that in the text.

NLL as an evaluation metric
NLL is used in the evaluation to compare the quality of generations, yet the results presented in Fig. 3 look a bit suspicious, because of the high variance within the subsequent tasks. Why is that?

One-class per task protocol
The authors evaluate their method under the assumption of disjoint classes par task. While this assumption is frequently used, it does not often correlate with the real-life conditions. Furthermore, the proposed method does not seem to provide any countermeasures against different splits. Following:
[Tzu-Ming Harry Hsu, Hang Qi, and Matthew Brown. Measuring the effects of non-identical data distribution for federated visual classification. arXiv preprint arXiv:1909.06335, 2019]
I believe the authors should evaluate their method using non-uniform and more complex class splits per task. This validation should provide a more thorough analysis of the proposed method.

Minor comments:

114 the following (approximated) optimal prior: (approximated or optimal - we cannot have both approximation and optimal solution

118 h looks like a density function, yet this symbol is introduced ad hoc without additional conditions for this function in (8), the notation should be fixed - does P
mean a set of density functions? It is unclear to me.

125 - 126 To this end, we add the entropy regularization - motivation missing

157 Train new component - it is unclear why h is part of R, not P as in the following equation.

190 gaussian -> Gaussian

212 empirically evaluation - empirically evaluate

Fig. 4 - keep -> keeps

a/the - please double check the work for the article corrections (e.g. 162 a/the training dataset

Naming in figures - the authors use "Boo" to describe their method, instead of BooVAE as introduced in the text. Why is that? I think it should be consistent.

**Time Spent Reviewing:**

4

---

> ### Author Response · Authors · 2021-08-11
> **Answering questions and incorporating feedback: reviewer 1sJx**
>
> We thank reviewer 1sJx for the positive and thoughtful feedback! We are encouraged that the reviewer finds the concept of leveraging pseudo-inputs to approximate an optimal prior fresh and likes the relationship with AdaBoost. We will incorporate the feedback and provide informal intuition about approach in L92-97 in the final version. We address specific questions below.
>
> [1sJx] *"I miss the motivation of focusing on VAE in terms of solving for continual learning. The proposed approach relates only to a single generative model, and it is not straightforward to me how to generalize this method to other generative models that can be used in generative replay, e.g. GANs."*
>
> 1.  VAE is a fairly popular generative model, which was shown to perform better or comparable to the state-of-the-art generative models. Hence we suppose using VAE  is not a rare setting. Our interest in this model is the following: real-world tasks are usually related to unlabeled and continually evolving data. Hence, we are interested in the methods of unsupervised and continual learning. The VAE model is attractive because of its ability to bind the data by the encoder with meaningful latent representations. This is an important feature for such practical applications as Rec.sys and Anomaly detection.
> Hence we propose the method designed with care to the structure of the model. We will update the Abstract in L1-3 and Introduction in L21-22 to make this motivation more clear.
> 2.  Our approach could be applied to generative models which share encoder-decoder structure with VAE, i.e. modification of the VAE with perceptual and adversarial losses. Application for pure GAN model is out of the scope of this paper.
> ---
> [1sJx] *Dataset selection*
> 1. *"I would recommend extending the dataset pool with Omniglot"*
>
> We evaluate the method broadly on MNIST, notMNIST, fashionMNIST and CelebA datasets and metrics (NLL, FID, Diversity). In Supp B.1. We provide per-task metrics changes as new tasks are arriving. Hence, we suppose that adding OMNIGLOT to this evaluation will not provide any additional insight.
>
> 2. *"the dataset changes (for instance train on MNIST, validate on FashionMNIST, etc. This becomes a standard of continual learning evaluation pipeline and gives more visibility in the algorithm performance."*
>
> We agree that continual learning is a broad term and switching between domains is a reasonable strategy, as a data-transfer problem. However, we focus on the particular setup of the same domain with new classes. Our arguments are the following:
> 1.  Concurrent works use one class per task protocol. We find this setup frequent for both generative and discriminative models
> 2.  As we consider static architecture it is reasonable to consider staying in the same data domain, but with new classes arriving: digits, people with the new hair color, new letters, etc. We are convinced that it is a practical set-up.
>
> 3. *"One-class per task protocol The authors evaluate their method under the assumption of disjoint classes par task. While this assumption is frequently used, it does not often correlate with the real-life conditions.<...>"*
>
> This is true for all the experiments except the CelebA. In this experiment, we construct unbalanced classes for different hair colors, as we discuss on L279-280.  Our approach performs well. We suppose this is due to the structure of the functional gradient, see Eq.12,13.
> For other datasets, we indeed followed frequently used assumptions. We agree that non-iid cases of data distributions are important and will discuss it and cite [Tzu-Ming Harry Hsu, Hang Qi, and Matthew Brown. Measuring the effects of non-identical data distribution for federated visual classification. arXiv preprint arXiv:1909.06335, 2019] in the related work section.
> ---
> [1sJx] *'"One of the requirements formulated is related to a generative replay and it claims that the generative replay methods need to assess (there's a missing 's') samples quality and task-balance, which leads to a computational overload. This requirement ends with the claim that the proposed approach should avoid generative replay. If I understand correctly this concerns the learning of a VAE (not a general continual learning setup), as in the evaluation section the authors use generative replay as a baseline. This should be clarified."'*
>
> In Introduction, we formulate requirements for a new method of continual VAE learning. We believe that training an additional generative model (as in generative replay) is not desirable in this setting. On the contrary, having a method for continual VAE learning allows us to use this model for continual learning of a discriminative model with reduced computational complexity (we do not need to train a generative model for each new task). We show in Sec. 5.2, how BooVAE performs in this setup.
>
> ---
> [1sJx] *"ELBO Maximization Step The authors introduce the function that allows to train VAE parameters, but it remains unclear if the ELBO is correctly defined here. The authors claim that they introduce a symmetric KLD, yet I'm unsure if it doesn't conflict with the lower bound."*
>
> 1.  Let’s start from an observation: in Eq.17 we subtract the sum of KL-div from the standard VAE ELBO for the task $t$ with data $\mathcal{D}^t$ L, which are non-negative. Hence, Eq.17 is still a lower bound. Please note that we are interested in optimization of the ELBO over the whole observed data $\mathcal{D}^{\leq t}$ and this is only ELBO for the task $t$.
>
> 2.  Next, we need to answer the question: is this lower bound still variational with respect to the $q_{phi}(\vec{z}|vec{u})$ or which additional gap is introduced? In the ELBO-part we have expectation over data $x\sim\mathcal{D}^t$ and in the regularization over learned pseudo-inputs $u$. Hence, the effect is similar to the amortization gap, introduced at Cremer C., Li X., Duvenaud D. Inference Suboptimality in Variational Autoencoders //arXiv e-prints. – 2018. – С. arXiv: 1801.03558. So, in practice, it should be even less than in case when we learn in the non-continual setting, and hence amortize all data with $q_{phi}(\vec{z}|vec{u})$.
>
> 3.  Finally, let us note that since the objective is still a lower bound, the regularization could be viewed in the same spirit as $\beta$-annealing or weight-decay.
>
> Thank you for your question, we will elaborate on this in Supp.
>
> ---
> [1sJx] *"Minorization-Maximization algorithm The authors use this term as if it is a well known method, while it seems to be the one introduced in this work. Please clarify that in the text."*
>
> MM is a popular family of optimization algorithms (Kenneth Lange: "MM Optimization Algorithms", SIAM,  (2016)). For example, EM algorithm, which is frequently used in machine learning, is also an example of MM algorithm. We will add the citation to the paper to avoid confusion. Thank you for pointing.
>
> ---
> [1sJx] *"NLL as an evaluation metric NLL is used in the evaluation to compare the quality of generations, yet the results presented in Fig. 3 look a bit suspicious, because of the high variance within the subsequent tasks. Why is that?"*
>
> This is an important observation. We discuss this in Supp.B.1 L521-529, Supp.B.2. We believe that these jumps could be attributed to the (dis-)similarity of some classes. E.g. in Fashion MNIST Task 6 is the first time the model sees shoes (class `Sandals`). Due to catastrophic forgetting, NLL on all the previous classes (`Coats`, `T-shirts`, etc.) becomes extremely large. Task 7, on the contrary, again deals with the more ‘similar’ class (`Shirts`), which lowers the NLL on all the previous classes (except for the `Sandals`) regardless of the presence of the forgetting. This is supported by Fig.8 and Fig.12, where we report per-task metrics and samples. Since BooVAE can deal with catastrophic forgetting, we do not observe large changes in NLL as the number of tasks increases.
>
> ---
> [1sJx]*"118 h looks like a density function, yet this symbol is introduced ad hoc without additional conditions for this function in (8), the notation should be fixed - does P mean a set of density functions? It is unclear to me."*
>
> This is correct. We introduce $\mathcal{P}$ in L120 as a set of probability densities. We will re-write these lines. Thank you for pointing this out.
>
> ---
> [1sJx]*"125 - 126 To this end, we add the entropy regularization - motivation missing"*
>
> We propose to approximate the optimal prior using a greedy procedure. On each step, prior is expanded by the projection of the functional gradient. The entropy regularization is a technical requirement for the well-posed projection problem and a nice choice for space of densities. Otherwise, since the problem is linear over $h$ (as scalar product of $h$ and functional gradient), the solution will be just delta function at the maximum point of $\alpha\frac{\hat{q}_1^{a}(z)}{\pi^{1}_{lambda}(z)} + (1-\alpha)\frac{\hat{q_2(z)}}{\pi^{1}_{lambda}(z)}$
>
> ---
> [1sJx]*"157 Train new component - it is unclear why h is part of R, not P as in the following equation."*
>
> $\mathcal{P}$ is a set of all probability densities. To perform optimization we need to choose the parametrization for h. In L150 we chose a particular family with pseudo-inputs to the encoder and named it $R$. We will simplify this and use $R$ everywhere as some parameterized family of densities.
>
> ---
> [1sJx]*"Naming in figures - the authors use "Boo" to describe their method, instead of BooVAE as introduced in the text. Why is that? I think it should be consistent."*
>
> Thank you for pointing out this inconsistency. We’ve skipped the “VAE” in the method label in the figures since all the methods in the comparison are using the VAE model. We will correct that to make the text and the plots consistent.
>
> We are thankful for your broad feedback! We are ready to add any further comments to clarify. We will update the manuscript accordingly, including all the minor comments that were kindly listed in the review.

---

> > ### Comment · Reviewer_1sJx · 2021-08-25
> > **Rebuttal response**
> >
> > Thank you for your response. I have updated my score accordingly.

---

> > > ### Author Response · Authors · 2021-08-26
> > > **Score update**
> > >
> > > Thank you for the time you spent with the review and the rebuttal! We note that despite your updated score, we don't see changes. We are not sure if it is a normal behaviour or something went wrong. We kindly ask you to check and also put the new score in the comments. We hope you understand our worries.

---

> > > > ### Comment · Reviewer_1sJx · 2021-08-26
> > > > **Score update**
> > > >
> > > > My final score is 6 and I see it correctly in the system. I hope this helps.

---

### Official Review · Reviewer_w1MQ · 2021-07-16

**Rating:** 8
**Confidence:** 3

**Summary:**

The authors propose a novel approach to tackle catastrophic forgetting. In more detail, they focus on variational autoencoders (VAEs) and investigate the role of the prior in the aforementioned setting. This work -- which is built on top of (Tomczak & Welling, 2018) that introduces a learnable prior through pseudo-inputs -- provides a definition of “optimal prior” in continual learning as well as an algorithm to update it when new tasks arrive. The authors evaluate the proposed approach on image generation (MNIST, its variants, and CelebA) and, moreover, as a generative memory that mitigates the forgetting of a discriminative model.

**Limitations And Societal Impact:**

See above.

**Main Review:**

I feel that this is a valid and strong submission:
- The motivations and the ideas are clear and well-stated.
- The paper remarkably differs from the current trend, which mainly focuses on classification.
- The approach appears grounded from a theoretical perspective.
- I appreciated the writing style: the section on Related work is concise and well-shaped; the conclusions lack but the reader does not miss it at all.
- The experimental evaluation is extensive and well-structured: the setups and protocols are clear; competitors and baselines adopted by the authors provide insightful suggestions and remarks to the reader; last but not least, the results are good.

My main concern regards Section 3, which I found particularly hard to follow. Honestly, this may partially be due to my limited knowledge of variational inference; however, I have a strong background in the closest related works (i.e. Vamp-Prior, VCL, and a plethora of CL methods). On the other hand, I think the authors should improve the readability of this section:
- Figure 1 is not insightful at all and does not provide an understanding of the approach.
- I read Section 3.2 several times, but still, I have only a vague idea about the functioning of the BooVAE algorithm.

**Time Spent Reviewing:**

3

---

> ### Author Response · Authors · 2021-08-11
> **Clarifications and incorporate feedback**
>
> We thank the reviewer w1MQ for the positive feedback! We are encouraged to read that the reviewer finds that the ideas differ from the current trend, are grounded from a theoretical perspective and are well-stated. We hope that our work will be useful. We address specific questions below and will *incorporate feedback* about readability in the final version.
>
> [w1MQ]
> 1. *"My main concern regards Section 3, which I found particularly hard to follow"*
> 2. *"I read Section 3.2 several times, but still, I have only a vague idea about the functioning of the BooVAE algorithm."*
>
> Thank you for the feedback. Let us provide more informal intuition. We start from Sec.3.2. We could note that maximization of the ELBO with respect to the parameters of the encoder, decoder is a valid step for any prior distribution, so we can encapsulate it. Also, maximisation of the ELBO with respect to the prior distribution only tightens the gap between lower bound and marginal loglikelihood. In Sec.3.1 we derive the solution for this problem: new optimal (argmax-ELBO) prior after the arrival of the new data. Since we know the unique optimal solution, we can minimize the discrepancy between the current prior and this optimum. Due to the additive nature of the solution, we also use recurrent additive perturbation of the current prior. Then, we could alternate between these two steps: ELBO maximization and prior update.
>
> We will improve the text by:
> 1. Provide more intuition at L92-97 at the very beginning of the text.
> 2. In both 3.1 and 3.2 we will lighten the notation, where possible: remove $\lambda$ index, use colour-code for different tasks, merge $\mathcal{P}$ and $\mathcal{R}_{u}$  between Sec. 3.1, 3.2 and other. In Sec. 3.1 we will merge Eq.8,9. In Sec. 3.2 we will add an example of learning two components for the approximation of an optimal prior after observing a new task.
>
> ----
> [w1MQ] *"Figure 1 is not insightful at all and does not provide an understanding of the approach."*
>
> We appreciate the feedback. In Fig. 1 we would like to show the relationship between data space and latent space: which objects we learn in dataspace and what is their influence (via prior and encoder) on the latent space. We will incorporate feedback in the following way:
> 1. Cut the "Task 1" subfigure and leave only the "Task $t$" part, also a bit compressed.
> 2. In the freed up space we will put a picture of learning 2D data with 1D latent space: noisy points around the line and colour code old points, old aggregated posterior density levels and functional gradient.
> We hope that this will make the illustration more insightful. Please, let us know if you have further feedback in mind, we will be really happy to incorporate it.
>
> ---
> We are thankful for your feedback. We are ready to add any further comments to clarify. We will update the manuscript accordingly.

---

### Official Review · Reviewer_3uR8 · 2021-07-16

**Rating:** 6
**Confidence:** 4

**Summary:**

This paper proposes a new approach for the continual learning problem of VAE. The key idea is to use trainable pseudo-inputs [Tomczak & Welling, 2018] to approximate the prior distribution of previous tasks using an entropy-based regularization method [Egorov et al., 2019], and aggregate the prior distribution to match the data points of the new task. On this basis, the proposed BooVAE algorithm is broadly evaluated on multiple datasets, and shown to mitigate catastrophic forgetting without using multi-head VAE architecture and generative replay.

**Limitations And Societal Impact:**

Yes.

**Main Review:**

Pros: The paper presents a sufficient literature review. Experimental results (especially qualitative results) show that the proposed approach is effective in dealing with catastrophic forgetting and improving sample diversity.

Cons:
1. Although the empirical results are good, the technical novelty of this paper remains unclear to me. If I understand correctly, the proposed method can be viewed as a combination of the existing VampPrior method and the entropy-based regularization technique from [Egorov et al., 2019], which improves the efficiency and diversity of the learned prior components.
2. In Fig. 3, compared with Random Coreset, the proposed method is shown to only achieve remarkable improvement when integrated with VCL.
3. It would be good if the proposed approach can be compared with more recent and competitive baseline models, such as CURL (Rao 2019), both in performance and computational efficiency.

A minor concern: It is suggested that the authors add a Conclusion section at the end of the paper to further summarize the contribution.


**Time Spent Reviewing:**

5 hours

---

> ### Author Response · Authors · 2021-08-11
> **Answer on concerns and incorporate feedback**
>
> We thank reviewer 3uR8 for the positive and thoughtful feedback. We are encouraged that the reviewer finds the empirical results to be good, the approach to be effective in dealing with catastrophic forgetting and the literature review to be sufficient.  We address specific questions and concerns below. We will *incorporate* provided feedback in the final version: add the Conclusion section and elaborate more in Related work about novelty and relation with [Tomczak & Welling, 2018], [Egorov et al., 2019].
>
> [3uR8] *"1.  Although the empirical results are good, the technical novelty of this paper remains unclear to me. If I understand correctly, the proposed method can be viewed as a combination of the existing VampPrior method and the entropy-based regularization technique from [Egorov et al., 2019], which improves the efficiency and diversity of the learned prior components."*
>
> This is true that the paper uses techniques from [Tomczak & Welling, 2018] [1] and [Egorov et al., 2019] [2] However, we don't completely agree that the paper is just using a combination of the methods and would like to elaborate on this.
>
> ***Novelty in idea-level***
>
> 1. We consider the innovation induced by new data arriving in data space as a perturbation over the current prior distribution. As opposed to [2], where the authors approximate posterior in the parameter space.
> 2. We would like to perform regularization by linking dataspace and latent space, instead of weight-space regularization.
>
> ***Novelty in technical-level***
>
> To practically implement this idea, we need to do several technical steps:
> 1. We decompose the ELBO in a static and continual learning setting (Eq.5, details Supp.A.1) which leads to Eq.6,7. This is different from derivations at [1] and not related to the [2].
> 2. Next, we introduce recursive approximation and derive the optimal solution in that setting: approximated optimal prior Supp.A.1, L458. This derivation is also new (while using standard variational calculus) and relates aggregated posterior to the continual learning problem. As we obtain the global solution, we can minimize the discrepancy between the current prior and the optimal one. This is different from the approach at [1].
> 3.   We propose to solve Eq.7. with additive perturbation of the current prior, as it has a special additive structure. We suppose that the idea is to match the innovation of arriving new data by an additive expansion of the density is novel. It is similar to boosting approaches. From the technical side, the objective is different from [2]: a convex combination of two *reversed* KL. We prove the bi-convexity of the optimization problem and derive the functional gradient. Supp.A.1, L470. We would like to clarify that source of the diversity is a greedy expansion of the prior by a projection of functional gradient. The entropy regularization is a technical requirement for the well-posed projection problem and a nice choice for space of densities.
>
> Hence, we suppose that the overall sequence of steps to obtain the algorithm is not trivial and uses special properties of the VAE model. We will *incorporate* this feedback and enrich Sec.4, L192-194 to make it more clear.
>
> ---
> [3uR8] *"2.  In Fig. 3, compared with Random Coreset, the proposed method is shown to only achieve remarkable improvement when integrated with VCL."*
>
> We agree that random coreset has a low NLL score (for NLL lower is better). However, we show that NLL doesn't capture the ability of the model to generate images from all the tasks. The diversity metric  (Fig.2; Supp.B.1, Fig.8, Fig.4) shows degradation in sampling from each task after several steps, Supp.B., L537-539. In Supp.B.4 we study the size of random to corest (per task) to reach the same performance as BooVAE. The difference is more than in several times. The FID metric, Table 1. also captures this type of behaviour. We discuss this drawback of the NLL metric in more detail at Supp.B.1, L521-539.
>
> ---
> [3uR8] *"3.  It would be good if the proposed approach can be compared with more recent and competitive baseline models, such as CURL (Rao 2019), both in performance and computational efficiency."*
>
> 1.  The main contribution of (D. Rao et al, 2019)[1] is the data-driven model expansion, i.e. identifying when the data-stream is sufficiently new to add a *new task-specific head to the encoder*. Then [1] combats with forgetting by deep *generative replay*:
> "during training, the model alternates between batches of real data, with samples $x_{\text{data}} ∼ D$ drawn from the current training distribution, and generated data, with samples $x_{\text{gen}}$ produced by the previous snapshot of the model (with parameters $\theta_{\text{prev}}$)"
>    [1] don’t provide FID or NLL for generation in a continual setting. We compare a variety of datasets including CelebA with VCL + Multihead as an instance of dynamic expansion algorithm, but without self-replay.
> 2. We would like to highlight our broad evaluation of the method with comparison to coresets, weight-penalty methods and multiheads in terms of datasets (MNIST, notMNIST, fashionMNIST, CelebA) and metrics (NLL, FID, Diversity). In Supp B.1. we provide per-task metrics changes as new tasks are arriving. In Sec.5.2 we also evaluate the ability of our approach to serving as a continually trained generative model to combat forgetting in discriminative models.
> ---
>
> We thank the reviewer for the thorough feedback and hope that we've managed to answer all the questions. We are ready to add any further comments to clarify. We will update the manuscript accordingly.

---

> > ### Comment · Reviewer_3uR8 · 2021-08-25
> > **Thanks for the response**
> >
> > Thanks for the response! It solves most of my problems.

---

> > > ### Author Response · Authors · 2021-08-26
> > > **Thanks for the response**
> > >
> > > Thank you for your time! We are encouraged that we've managed to address your concerns.

---

### Official Review · Reviewer_T9Fg · 2021-07-17

**Rating:** 6
**Confidence:** 4

**Summary:**

This manuscript proposes a continual learning algorithm for VAE.  The encoder q(z|x) and decoder p(x|z) are apparently untouched when new tasks are learned; instead, continual learning here focuses on the prior pi(z).  The prior is adapted to approximate, as well as possible, the average of q(z|x) over all tasks observed until the present.  This is done by an algorithm similar to coreset selection, but instead of selecting data from each task, the algorithm learns representative vectors, u, from each task such that the average of q(z|u) matches the cumulative sum of q(z|x) as well as possible.

**Limitations And Societal Impact:**

Descriptions of the relationship of this work to prior art overstate the potential impact of this work.  This is especially the case on page 1: (1) The manuscript tries to say that learning a new head for each new task is theoretically impossible because the latent space should not change, but since Nguyen et al. have shown that multi-head continual VAE works pretty well, this argument seems self-defeating. (2) The manuscript tries to say that generative replay is too expensive, but since generative replay can be accomplished by something as simple as sampling from a memorized coreset, this argument also seeems self-defeating.



**Main Review:**

Strengths

The idea is quite innovative.  It is a significant improvement over coreset selection, the algorithm that it most resembles.

Weaknesses

Why is it reasonable to update only the prior?  Why is it not necessary to also update the encoder and decoder in order to represent new tasks?

If there is a new set of U vectors selected for each new task, then it seems that space complexity grows linearly with the number of tasks observed.  This problem is also suffered by coreset selection and by multi-head VAE, but not by VCL; since VCL performs so well in your experiments, it would be interesting to compare their performance and their space complexity.

Eq. (10) is only correct up to O(beta).  Beta is not a noise term, it is the interpolation constant, so its size is important to the problem; why is ignoring O(beta) an acceptable approximation?

Eq. (11) is true if the denominator is \pi_\lambda^{1,2}(z), because of Eq. (7), but it is not true if the denominator is \pi_\lambda^{1}(z) as shown.

The notation is difficult to follow; there are so many different versions of the prior, and some of them seem to mean exactly the same thing.  For example, if r^{t}(z)=\pi_\lambda^{t}(z), then why is the r^{t}(z) notation needed?  Indeed, why use h(z) instead of \pi_\lambda^{2}(z)?  With all of these variables floating around, a table listing the variables would be useful.

There are frequent minor English mistakes, but they do not obscure understanding .  For example, "asses" -> "assess," "Maximization" capitalized in the middle of a sentence, "to it approximation", "additive expansions" -> "additive expansion", "this reached."



**Time Spent Reviewing:**

4

---

> ### Author Response · Authors · 2021-08-10
> **Answer on questions and incorporate feedback**
>
> We thank the reviewer T9Fg for positive and thoughtful feedback. We are encouraged that the reviewer finds the idea quite innovative and imporovement over coreset selection significant.  We address specific questions below and will *incorporate all feedback* in the final version.
>
> [T9Fg] *"The encoder $q(z|x)$ and decoder $p(x|z)$ are apparently **untouched** (1) when new tasks are learned; instead, continual learning here focuses on the prior $\pi(z)$".<...> This is done by an algorithm similar to coreset selection, but instead of selecting data from each task, the algorithm learns representative vectors, u, from each task such that the average of $q(z|u)$ matches **the cumulative sum of $q(z|x)$** (2) as well as possible.*
>
> We would like to comment on (1), (2) statements to avoid misunderstanding.
> 1. (1) Continual learning indeed focuses on the prior, but parameters of the encoder, decoder are also adjusted as new tasks are coming. We present this in Sec.3.2 (L134-142). In the first step (L143), we update the prior to match new aggregate posterior, while the second step is ELBO Maximisation (L173), where parameters of the encoder and decoder are updated, see Alg.1.
> 2. (2) This is correct: our prior aims to match **the cumulative sum of $q(z|x)$** (2) **over all the tasks**. But for a single new task an algorithm involves more than just approximation of sum $q(z|x)$, $x\in\text{new task}$. We match innovation induced by new data relative to the already aggregated, Eq.12,13.
> ---
> [T9Fg] *"Why is it reasonable to update only the prior?"*
> We update also parameters $\phi,\theta$. Please see Sec.3.2: L134-142, Alg.1.
> ---
> [T9Fg]
> 1. *it seems that space complexity grows linearly with the number of tasks observed.*
>
> Indeed, with each new task we need to add less or equal than budget new vectors $u$. However, in Supp.B.4 we study the size of random to corest (per task) to reach the same perfomance as BooVAE. The difference is more that in several times, hence in practice the computational complexity is different.
>
> 2. *VLC performes so well in your experiments*
>
> We do not completely agree with this statement. The NLL metric is indeed comparative. However, the diversity metric  (Fig.2; Supp.B., Fig.8) shows degradation in sampling from each task after several, Supp.B., L537-539. The most illustrative are samples after each task in Fig.10-12. This is also captured by FID on CelebA, Table 1.
>
> ---
> [T9Fg] *"Eq.10 is only correct up to $O(\beta)$. $\beta$ is not a noise term, it is the interpolation constant, so its size is important to the problem; why is ignoring $O(\beta)$ an acceptable approximation?"*
>
> 1. Please note it isn't $O(\beta)$, but $o(\beta)$: $\lim_{\beta\to 0}\frac{o(\beta)}{\beta}=0$, i.e. we consider only first-order term and drop $\beta^n, n>1$ to obtain functional *gradient*. We also provide details in Supp.A.1, L473, Eq.29.
> 2. Let us recall L120-121, L157-L160: learning prior is 2-step: 1. Find $h$ by the projection of the functional gradient to the selected density family $R$. 2. Optimize KL over $\beta$.  This strategy is valid, as an instance of Functional Frank-Wolfe algorithm (L118-119) applied to the bi-convex objective Eq.8. We show bi-convexity of the task at Supp.A.1, L464-469.
>
> Less formal: we greedily catch the novelity $h$ up to the first order of $\beta$, then select this "step-size" by optimizing over $\beta$ and repeat. The approach is greedy, hence is not a global optimal. At L171-172 and Supp.A.3 we comment on this.
>
> ---
> [T9Fg] *"For example, if $r^{t}(z)=\pi_\lambda^{t}(z)$, then why is the $r^{t}(z)$ notation needed?"*
>
> Let us clarify the difference:
> 1. $\pi_{\lambda}^{\leq t}(z)$ = current approximation
> ($\pi_{\lambda}^{\leq t-1}(z)$) + true aggegated posterior of the new task $t$  (Eq.14)
> 2. We start adding new components $h_k$ to $\pi_{\lambda}^{\leq t-1}(z)$, to "catch" the innovation induced by true aggregated posterior of the new task $t$.
> At this point $r^{t}$ from L149-156 and Alg.1 appears: $r_{0}^{t}=\pi_{\lambda}^{\leq t-1}(z)$, as we didn't yet added anything. Next, we add the first mixture component $r_{1}^{t}=(1-\beta_1)\pi_{\lambda}^{\leq t-1} + \beta_1h_1$.Then, the second mixture component: $r_{2}^{t}=(1-\beta_2)[(1-\beta_1)\pi_{\lambda}^{\leq t-1} + \beta_1h_1]+\beta_2h_2=(1-\beta_2)r_{1}^{t}+\beta_2h_2$.
> 3. Finally, when we reach the stopping criteria (Alg.1) and $\pi_{\lambda}^{\leq t}(z)=r_{K}^{t}$.
>
> Hence, $r_{k}^{t}$ on some iteration $k$ is the ongoing greedy approximation after observing the task $t$, and $\pi_{\lambda}^{\leq t}(z)$ is the result.
>
> See also L127-129. We agree: notation is complicated. This is due to the nested reccursion: 1. As new tasks arrives, the aggregated posterior is updated recurrsively with new data 2. Inside new task, approximation of the new prior is also reccursive. We will *incorporate* this feedback: clarify the notation by dropping $\lambda$, using color-coding and including table with notation.
>
> ---
> [T9Fg] *"Indeed, why use $h(z)$ instead of $\pi_\lambda^{2}(z)?$"*
>
> $h(z)$ is additive pertrubation of *current* approximation towards the target. $\pi_\lambda^{1,2}(z)$ (we didn't use  $\pi_\lambda^{2}(z)$) is the approximation of the aggregated posterior of the task 1 + true aggegated posterior of the task 2, see Eq.7.
>
> ---
> [T9Fg] *"Limitations And Societal Impact: Descriptions of the relationship of this work to prior art overstate the potential impact of this work. This is especially the case on page 1: <...>""*
>
> We consider our introduction only as the motivation for new approaches, with all respect to the work of other authors.
>
> 1. *"The manuscript tries to say that learning a new head for each new task is theoretically impossible because the latent space should not change, but since Nguyen et al. have shown that multi-head continual VAE works pretty well, this argument seems self-defeating.''*
>
> We agree with this statement in general, but we don't agree that we claim theoretical impossibility. Indeed, multi-heads perform well. Hence, we compare with it (Table 1, 2) and reach comparative or better perfomance, while using the static architecture (6 times less parameters), see Supp.B.1 for discussion. In L26 we only claim that: we *suppose*, that practical applications of VAE require common latent space for all task, hence it is better to avoid multiheads. We will *incorporate* this feedback: change from "should keep static architecture" to "we want to keep static architecture" in (L29). [we believe that it is benificial to have static architecture]
>
> 2. *"(2) The manuscript tries to say that generative replay is too expensive, but since generative replay can be accomplished by something as simple as sampling from a memorized coreset<...>."*
>
> We also agree with this in general, so we compare with memorized coreset. By *generative replay* we consider a particular approach of (re-)training generative model to reproduce distriubtion of training data, described in L32-35. We will *incorporate* this feedback by highlighting the differences with coresets.
>
> ---
>
> We again thank the reviewer for the thourough feedback and hope that we've managed to answer all the questions. We are ready to add any further comments to clarify and will update the manuscript accordingly.

---

### Decision · Program_Chairs · 2021-09-27

**Decision:**

Accept (Poster)

**Comment:**

The submission proposes a learnable prior for continual learning of a VAE to retain performance on previous tasks.  The reviewers were unanimous that the paper is above the threshold for acceptance at NeurIPS.  Quoting from Reviewer T9Fg "The idea is quite innovative. It is a significant improvement over coreset selection, the algorithm that it most resembles."  The reviewers also appreciated both the theoretical and empirical results in the paper.  From Reviewer w1MQ "The experimental evaluation is extensive and well-structured: the setups and protocols are clear; competitors and baselines adopted by the authors provide insightful suggestions and remarks to the reader; last but not least, the results are good."  Some concerns remained, including some language typos, and clarity of presentation (comments by T9Fg, w1MQ - section 3, and 1sJx).